# BooookScore:
# A systematic exploration of book-length summarization in the era of LLMs

**Yapei Chang**
University of Massachusetts Amherst
yapeichang@umass.edu

**Kyle Lo**
Allen Institute for AI
kylel@allenai.org

**Tanya Goyal**
Princeton University
tanyagoyal@princeton.edu

**Mohit Iyyer**
University of Massachusetts Amherst
miyyer@cs.umass.edu

## Abstract

Summarizing book-length documents (>100K tokens) that exceed the context window size of large language models (LLMs) requires first breaking the input document into smaller chunks and then prompting an LLM to merge, update, and compress chunk-level summaries. Despite the complexity and importance of this task, it has yet to be meaningfully studied due to the challenges of evaluation: existing book-length summarization datasets (e.g., BookSum) are in the pretraining data of most public LLMs, and existing evaluation methods struggle to capture errors made by modern LLM summarizers. In this paper, we present the first study of the *coherence* of LLM-based book-length summarizers implemented via two prompting workflows: (1) *hierarchically merging* chunk-level summaries, and (2) *incrementally updating* a running summary. We obtain 1193 fine-grained human annotations on GPT-4 generated summaries of 100 recently-published books and identify eight common types of coherence errors made by LLMs. Because human evaluation is expensive and time-consuming, we develop an automatic metric, BooookScore, that measures the proportion of sentences in a summary that do not contain any of the identified error types. BooookScore has high agreement with human annotations and allows us to systematically evaluate the impact of many other critical parameters (e.g., chunk size, base LLM) while saving $15K USD and 500 hours in human evaluation costs. We find that closed-source LLMs such as GPT-4 and Claude 2 produce summaries with higher BooookScore than those generated by open-source models. While LLaMA 2 falls behind other models, Mixtral achieves performance on par with GPT-3.5-Turbo. Incremental updating yields lower BooookScore but higher level of detail than hierarchical merging, a trade-off sometimes preferred by annotators. We release code and annotations to spur more principled research on book-length summarization.

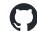 github.com/lilakk/BooookScore

## 1 Introduction

Just two years ago, automatically-generated summaries were riddled with artifacts such as grammar errors, repetition, and hallucination (Zhao et al., 2020; Fabbri et al., 2020; Goyal & Durrett, 2021). Nowadays, such artifacts have mostly disappeared; in fact, Pu et al. (2023b) find that summaries generated by large language models (LLMs) are preferred over those written *by humans*, leading them to pronounce the death of summarization research. However, as with most prior work on summarization, the input documents in their study are relatively short (<10K tokens). Widespread adoption of LLMs outside the research community has driven the development of a more ambitious task: summarizing *book-length* documents, which we define to be texts longer than 100K tokens.

As these documents exceed the context window limits of today's LLMs (e.g., 8K tokens for GPT-4), summarizing them via prompt-based approaches necessitates heuristics to chunk the input, process each chunk, and then combine and compress the outputs (Wu et al., 2021).

Despite the promise that LLMs hold for long-context tasks, the research community still lacks a principled and systematic approach to evaluate their capabilities on book-length summarization. Our paper identifies three open challenges with evaluation: (1) *data contamination*, in which existing benchmarks such as BookSum (Kryscinski et al., 2022) are in the pretraining data of modern LLMs (Chang et al., 2023); (2) *an unexplored error distribution*, as most prior summarization research centers around short source documents and fails to capture coherence errors that are exacerbated by the "chunk and combine" book-length summarization setting; and (3) *a lack of any reliable automatic metric*, which requires careful design and validation against human annotations.

**Contribution 1: A protocol for evaluating coherence in book-length summarization (§3).** To mitigate the impact of data contamination, we design our evaluation framework around the use of *newly-published books*. We propose a reference-free protocol that leverages human annotation of the *coherence* of LLM-generated summaries (i.e., their logical connectedness) under different prompting strategies. Our protocol unifies and extends best-practices across disparate works in document understanding and evaluation research, including adoption of fine-grained annotation units, use of QA pairs to denote points of confusion, and a taxonomic breakdown of different coherence errors.

We validate our protocol by collecting 1193 span-level human annotations on GPT-4 generated summaries of a carefully curated set of 100 recently-published books (costing $3K USD and 100 annotator hours) using two prompting strategies (hierarchical merging and incremental updating, shown in Figure 1). In categorizing these annotations into eight frequent error types, we reveal an error distribution in GPT-4 summaries that differs from that observed in prior studies on short-document summarizers; notably, we identify new error types (causal omissions, salience errors) through our book-length summarization setting (Table 1).

**Contribution 2: An automatic metric—BoooookScore—to assess summary coherence (§4).** Since our human evaluation is expensive, we follow recent work by developing an LLM-based evaluation metric called BoooookScore that identifies and explains instances of any of our eight established coherence errors in a given summary. Human validation shows that BoooookScore's annotations are almost as reliable as those of human annotators, which allows us to automatically evaluate many other book-length summarization configurations. Because BoooookScore does not rely on gold summaries, it can easily be used to evaluate new LLM summarizers on any collection of newly-published books, ensuring that the metric will remain meaningful for LLMs of the future.

**Contribution 3: A systematic evaluation of different LLMs using BoooookScore (§5).** We use BoooookScore to evaluate the impact of several critical design decisions on the coherence of generated summaries, including the choice of prompting strategy, base LLM, and chunk size, a study that altogether cost $10K (USD) in LLM API calls. Our findings include (1) hierarchical merging generally results in more coherent summaries but reduced level of detail compared to incremental updating; (2) GPT-4 and Claude 2 produce the most coherent summaries, while LLaMA 2 is substantially worse and fails to follow instructions; (3) increasing the chunk size does not improve hierarchical merging but does substantially benefit Claude 2 when using incremental updating; and (4) summary-level preference judgments are highly subjective and do not correlate with BoooookScore.

## 2 BACKGROUND: SUMMARIZING BOOK-LENGTH TEXTS WITH LLMS

Before discussing our evaluation protocol, we first outline two strategies—*hierarchical merging* and *incremental updating*—for prompting an LLM to summarize book-length documents that exceed its maximum context size. In both strategies, the length of the input document necessitates first dividing it into smaller chunks and then repeatedly merging, updating, and/or compressing chunk-level partial summaries (Figure 1). While neither strategy is well-explored by published research, hierarchical merging essentially adapts the strategy proposed by Wu et al. (2021) to zero-shot prompting, while incremental updating resembles chain-of-density prompting proposed for short-document summarization (Adams et al., 2023). Both are implemented in widely-used open-source LLM libraries such as LangChain,[1] but the relative merits of each method remain unexplored.

---

[1] LangChain implements incremental updating via refine and hierarchical merging via map-reduce.

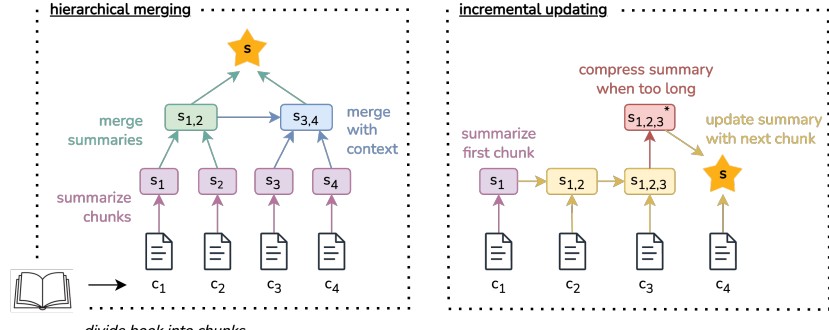

Figure 1: To perform book-length summarization, we first divide a book into smaller chunks that fit within the context window of an LLM. Then, we explore two strategies for summarization: (1) *hierarchical merging*, in which chunks are first summarized and then the corresponding summaries merged via separate prompts; and (2) *incremental updating*, in which a global summary is updated and compressed as we step through the book chunk-by-chunk.

More specifically, both strategies assume an LLM with context window size $W$ is used to summarize an input document $D$ whose length $L \gg W$. We thus split $D$ into non-overlapping chunks $c_1, c_2, \ldots c_{\lceil \frac{L}{C} \rceil}$ where $C < W$ is the length of each chunk.[2]

**Hierarchical merging:** Wu et al. (2021) propose a method in which an LLM (in their case, GPT-3) is fine-tuned via reinforcement learning to summarize each chunk and then hierarchically merge the chunk-level summaries until one summary is left of the entire input document. This method has since been simplified into a zero-shot prompting strategy without further training, as shown in Figure 1 (left). Hierarchical merging requires three unique prompts for (1) summarizing an input chunk, (2) merging chunk-level summaries, and (3) merging summaries with added context from previously-generated merged summaries. We ensure that the total length of each prompt and its associated inputs is less than $W - G_l$, where $G_l$ is a hyperparameter controlling summary length that varies depending on the level $l$. Summaries are recursively merged until only one summary (of the full book) remains; see Appendix A.1 for further details.

**Incremental updating:** It is possible that since hierarchical merging necessitates summarizing portions of the input document without complete context, it may introduce more coherence errors. For example, in the first level, chunks towards the end of the book will be summarized without knowledge of what came before, which can lead to incoherent summaries especially for non-linear or multi-perspective narratives. We thus explore an alternate prompting strategy—*incremental updating* (Figure 1, right)— that iterates through each chunk in order while continuously updating a global summary with salient information. While this method may be better able to handle inter-chunk dependencies than hierarchical merging, it requires more complicated prompts for (1) summarizing an input chunk, (2) updating the global summary $s_{1,2,\ldots,i-1}$ with information from the current chunk $c_i$, and (3) compressing the global summary when it exceeds the maximum summary length $G_n$. See Appendix A.2 for a full specification of incremental updating.

## 3  EVALUATING COHERENCE OF BOOK SUMMARIES

In this section, we define our framework for human evaluation of coherence errors in book-length summarization. Our framework involves: (1) corpus collection focusing on newly-published books, (2) unification and extension of best-practices from prior document understanding and evaluation literature to guide data annotation, and (3) analysis of human annotations centered around emergent coherence error categories of summaries generated by modern LLMs.

---

[2]We ensure each chunk ends at a sentence boundary.

**Collecting a corpus of newly-published books.** The only existing public dataset for book-length summarization is BookSum (Kryscinski et al., 2022), which contains famous books from the Project Gutenberg public-domain repository along with reference summaries scraped from popular websites such as CliffNotes and GradeSaver. Both the source books and reference summaries are in the pretraining data of existing LLMs: Chang et al. (2023) confirm that many books in the BookSum held-out split (e.g., *The Adventures of Huckleberry Finn*, *The Picture of Dorian Gray*) are among the most-memorized books by GPT-4 and GPT-3.5-Turbo, and we were able to auto-complete several reference BookSum summaries by prompting GPT-4 with a short prefix of the summary.

To reduce the confounding impact of summary memorization, we manually collect 100 books[3] published within the past year to form our dataset (see Table 3 for a full list). Some of these books could still have appeared in the pretraining dataset of recent LLMs such as Claude 2 and LLaMa2, although it is much less likely than in BookSum. However, *summaries* of these books do not publicly exist: we did not find *summaries* online for any books in our dataset, which significantly lowers the possibility of LLM memorization.[4] The average length of the books in our dataset is 190K tokens, compared to 112K tokens in BookSum. Due to copyright laws, we cannot publicly release this dataset; even if we could, we would still recommend that researchers collect their own datasets of newly-published books to minimize contamination with LLMs of the future.

**An evaluation framework for book-length summarization.** Since we lack gold summaries, we design our evaluation framework to be reference-free, which aids in scalability. To do this, our evaluation framework synthesizes best-practices of prior document understanding and summarization evaluation research. Our evaluation employs: (1) *fine-grained evaluation units* as recommended by LongEval (Krishna et al., 2023); (2) information-seeking questions to represent naturally-occurring *points of confusion* (Ko et al., 2020; Wu et al., 2023; Meng et al., 2023; Newman et al., 2023); and (3) focus on *summary coherence*, which evaluates the logical structure and readability of the summary itself (Goyal et al., 2022a). We do not directly evaluate the *faithfulness* of the summaries (i.e., how factually accurate they are at conveying information from the source text), as the length of the source texts poses considerable issues for any existing faithfulness evaluation. We qualitatively discuss faithfulness in Section 5 and leave further investigation for future work.

**Annotation protocol.** We implement our framework through a source- and reference-free annotation protocol where (1) annotators read through an LLM-generated summary, (2) highlight all confusing spans, and (3) ask question(s) for each marked span that highlight their confusion.[5] See Table 1 (third column) for examples of spans and questions produced by our annotators. We hired four annotators with extensive English proofreading experience on Upwork[6], each of whom annotated 25 disjoint summaries. Each summary takes roughly 30 minutes to fully annotate with spans and questions, and we paid $15 USD per summary for a total of $3K to evaluate both prompting strategies. To generate the summaries, we set the base LLM to GPT-4 with a chunk size of 4096 and a maximum summary length $G_n = 1200$; other hyperparameters are detailed in Section 5. In total, the annotators mark **840** (incremental updating) and **353** (hierarchical merging) coherence errors for GPT-4-generated summaries; see Table 1 (right) for the split across error types.

**Validating the annotations:** Typical measures of agreement are difficult to obtain in our setup, as measuring recall would require ground truth annotations with all possible coherence errors in the summaries; additionally, Goyal et al. (2022a) and Dou et al. (2022) observed low recall among annotators when evaluating machine-generated text at a fine-grained level. This motivates us to instead measure the *precision* of a given error annotation (i.e., after reading the corresponding question, do you agree that the span is confusing?), as it is simpler and cheaper while still being an informative metric. Given a span from a summary marked as containing an error, along with questions highlighting the confusion, we ask annotators (1) whether they think the span is confusing; and (2) whether the corresponding questions highlight the central confusion. We use the same four annotators hired before for this task, but make them validate human and (and later GPT-4) annotations for 25 books that they did *not* annotate in the first task. Overall, we validated 1,659 annotations for a total cost of

---

[3]We roughly balance our dataset across the following genres: fiction, non-fiction, sci-fi, fantasy, historical, contemporary, and memoir. We also include both linear and non-linear (multi-perspective and time-shifting) narratives in the dataset, and we purchase electronic copies of each of the 100 books in the dataset.

[4]However, we did find book reviews, which intentionally do not reveal major plot points or other spoilers.

[5]We also enabled forming relations between two spans in case multiple spans contributed to the same issue.

[6]http://upwork.com

Table 1: Definition of all coherence error types, an example annotation for each, and their prevalence (%) in generated summaries, which is calculated as the number of error occurrences in all summaries normalized by the total number of sentences in all summaries.

| Error Type | Definition | Example spans & questions | % errors per sentence inc / hier |
|---|---|---|---|
| Entity omission | An entity (e.g., person, object, place) is mentioned in the summary, but key context or details are missing or unclear. | *Span*: A mysterious man introduces Proctor to "Arrivalism." *Question*: Who is this mysterious man? | 7.3 / 3.71 |
| Event omission | An event is mentioned in the summary, but key details are missing or unclear. | *Span*: During a mission to find Caeli, Proctor is captured by watchmen while Thea escapes. *Question*: What happened to Caeli? | 4.25 / 2.27 |
| Causal omission | A reason or motivation is missing or under-explained. | *Span*: Proctor seeks answers from... Callista about the investigation. *Question*: Why would Callista know something about the investigation? | 2.75 / 1.21 |
| Discontinuity | An interruption in the flow of the narrative such as sudden jumps in time or perspective. | *Span*: In the new settlement, Thea adjusts to her life, working hard and finding solace in nature. *Question*: Why the shift to Thea's perspective? | 2.23 / 1.56 |
| Salience | Inclusion of details that do not contribute to the main plot. | *Span*: His father... flees, resulting in a chaotic chase on the pier. *Question*: What is the significance of this incident? | 1.42 / 1.03 |
| Language | Spelling or grammar issues; ambiguous wording. | *Span*: Despite her love for him, Deborah is heartbroken by his decision. *Question*: Why is the preposition "Despite" used here when she is, in fact, heartbroken because of her love for him? | 0.82 / 0.71 |
| Inconsistency | A discrepancy or contradiction within a story's plot, character development, or themes. | *Span*: In a farewell, Proctor marries his brother Malcolm to Cynthia and says goodbye to his loved ones. *Question*: If Cynthia is his mother and Malcolm is his brother, how can a mother and son marry? | 0.97 / 1.03 |
| Duplication | Redundant repetition of similar information. | *Span 1*: Proctor... deals with students and school issues, seeking help from Callista to fund a roof replacement. *Span 2*: Proctor's life continues as he... deals with school issues, such as funding for a roof replacement *Question*: Why does the same information appear twice? | 2.12 / 1.18 |

$418.90 (USD),[7] and we discover that **79.7%** of annotated spans are validated as legitimate through this task. More details on our validation can be found in Appendix J.

**Categorizing coherence errors:** After collecting spans and questions from the annotators, we develop an error taxonomy consisting of the eight types detailed in Table 1, which covers the vast majority of annotations, and we manually code each annotation using this taxonomy. We intentionally went through this process without relying on the SNaC taxonomy (Goyal et al., 2022a) so as to not be overly influenced by their error annotation schema which was tailor-made for fine-tuned summarization models. While we find considerable overlap in the two error schemas, we also discover two new instances of prominent errors not present in SNaC: *causal omissions* and *salience issues*. Our taxonomy also places less emphasis on language errors (e.g. coreference issues from SNaC) since modern LLMs rarely make such mistakes (Goyal et al., 2022b). Table 1 shows that omission errors are the most common across both incremental and hierarchical prompting strategies, and also that hierarchical merging makes fewer errors of every type but inconsistencies.

## 4 BOOOOKSCORE: AN AUTOMATIC EVALUATION METRIC

Since human evaluation of summary coherence is not scalable due to the high financial and time cost, we develop an automatic metric — BOOOOKSCORE— that prompts an LLM to identify instances of the eight error types we identified in Section 3. We validate BOOOOKSCORE via a human evaluation of its precision (following the annotation task discussed in the previous section) and show that its precision matches that of human annotators (78.2% vs. 79.7%). We then use BOOOOKSCORE to evaluate many other book-length summarization configurations, saving $15K USD in evaluation costs and 500 hours in annotator time. We emphasize that incorporating definitions and examples from our error taxonomy into the prompt is critical to achieve high precision with BOOOOKSCORE.[8]

---

[7] This cost includes validation of both human and BOOOOKSCORE annotations.

[8] In preliminary experiments without definitions and few-shot demonstrations, we qualitatively observe significantly reduced annotation precision.

## 4.1 IMPLEMENTING BOOOOKSCORE

Motivated by prior successful efforts to evaluate LLM-generated text via LLMs, such as AlpacaEval (Dubois et al., 2023), FActScore (Min et al., 2023), and G-Eval (Liu et al., 2023b), BOOOOKSCORE automatically measures the coherence of summaries generated by a book-length summarization system via few-shot prompting. BOOOOKSCORE is both source-free and reference-free (i.e., it does not require access to the input book or a reference summary), similar to the SNaC classifier built for fine-tuned summarizers by Goyal et al. (2022a).

**Specification:** Assume we have a summary $S$ consisting of sentences $s_1, s_2, \ldots, s_n$.[9] We develop a few-shot error-identification prompt $E$ that instructs the LLM to identify any instances of one of the eight specified error types in a given sentence $s_i$ of the summary. Concretely, we iterate over each sentence $s_i$ in the summary, feeding the prompt $E$, full summary $S$, and target sentence $s_i$ at each step. There are two acceptable outputs at each step: either (1) no error is found and the LLM outputs `No confusion`, or (2) an error(s) is identified and the LLM is asked to generate a corresponding question and associated error type. We include two full summaries with 42 sentence-level annotations in the prompt as demonstrations.[10] The BOOOOKSCORE of a single summary $S$ (Figure 2) is then computed as:

$$\text{BOOOOKSCORE}(S) = \frac{1}{n} \sum_{s_i \in S} [\text{LLM}(E, S, s_i) == \texttt{No confusion}] \tag{1}$$

When computing BOOOOKSCORE, we consider each sentence as a singular unit of confusion, rather than each of the questions associated with that sentence. This is because both LLMs and human annotators occasionally ask multiple questions that essentially target the same issue within a given sentence.[11] Thus, our metric intuitively measures the proportion of sentences in the summary that contain no errors (i.e., higher is better). To obtain a system-level score, we compute the mean BOOOOKSCORE across all summaries generated by that system.

**Validating BOOOOKSCORE:** We validate BOOOOKSCORE annotations in the same way that we validate human annotations in Section 3: by hiring human annotators to judge whether they agree with an LLM-generated annotation (here, GPT-4). We observe that the precision of human annotations is **79.7%**, while the precision of BOOOOKSCORE annotations is **78.2%** (details in Appendix J). Additionally, we compute BOOOOKSCORE using *human annotations* instead of LLM-generated ones for both GPT-4 configurations (i.e., replacing LLM($E, S, s_i$)

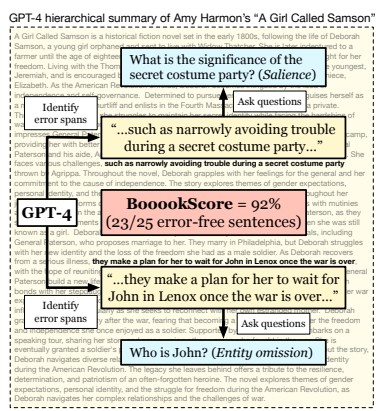

Figure 2: BOOOOKSCORE measures the proportion of error-free sentences in a summary, where coherence errors are detected by prompting GPT-4.

in Equation 1 with the human error annotation for $s_i$) and observe extremely similar system-level scores. Using human annotations in Equation 1 yields a BOOOOKSCORE of **82.1** and **89.4**[12] for

---

[9]After iterating over the design in numerous preliminary experiments, we find that our prompt works most reliably at the sentence level, rather than at the full summary level. As such, sentence tokenization is a required preprocessing step for BOOOOKSCORE. Future work should focus on implementations at the summary level, as it would save many calls to the LLM; here, we need to prompt the model separately for each sentence.

[10]These examples contain a combination of sentences with and without confusion, all the while maintaining a diverse range of error types. The full prompt can be found in M.4.

[11]For example, the questions "Who is John? Is he Lia's husband?" both seek to establish John's identity. Counting the number of questions instead of highlighted sentences would inadvertently overstate the weight of certain errors found within the same sentence.

[12]Recall that human annotators can (1) highlight multiples consecutive sentences as one span and (2) create relations between two spans, while GPT-4 can only highlight single sentences as spans. To adjust for this difference, we treat both consecutive sentences and relations as single sentences when computing BOOOOKSCORE for humans.

Table 2: BOOOOKSCORE for summaries generated under different configurations; higher scores indicate better coherence. We additionally report the *average summary length* in tokens based on tiktoken (https://github.com/openai/tiktoken) tokenizer, the *percentage of novel trigrams* compared to the source, and *percentage of repeated trigrams* in the summary.

| Model | Chunk size | BOOOOKSCORE | Avg. length | % novel 3-grams | % rep. 3-grams |
|---|---|---|---|---|---|
| *Summaries generated via hierarchical merging* | | | | | |
| GPT-4 | 2048 | 89.1 | 778.6 | 82.4 | 4.2 |
| GPT-3.5-Turbo | 2048 | 84.2 | 667.3 | 82.8 | 9.0 |
| Claude 2 | 2048 | 91.1 | 522.6 | 88.4 | 1.3 |
| Claude 2 | 88000 | 90.3 | 551.5 | 87.1 | 2.0 |
| Mixtral-8x7B | 2048 | 81.5 | 679.1 | 85.9 | 4.1 |
| LLaMA2-7B-Inst | 2048 | 72.4 | 684.9 | 76.4 | 36.1 |
| *Summaries generated via incremental updating* | | | | | |
| GPT-4 | 2048 | 82.5 | 805.4 | 84.1 | 3.4 |
| GPT-3.5-Turbo | 2048 | 67.0 | 484.5 | 68.2 | 3.5 |
| Claude 2 | 2048 | 78.6 | 657.1 | 89.4 | 1.9 |
| Claude 2 | 88000 | 90.9 | 493.7 | 84.7 | 1.9 |
| Mixtral-8x7B | 2048 | 64.5 | 558.7 | 82.3 | 3.5 |

GPT-4 summaries generated via incremental updating and hierarchical merging, respectively, while using LLM annotations yields a BOOOOKSCORE of **82.4** and **90.8**. Figure 4 compares the error distributions from GPT-4 to those of human annotators and shows that GPT-4 is more sensitive to omission errors and less sensitive to duplication or language errors. Taken as a whole, these results confirm that BOOOOKSCORE is a reliable annotator of coherence for book-length summarization. While we implement BOOOOKSCORE with GPT-4 for the remainder of this paper, implementing BOOOOKSCORE with open-source LLM annotators is an exciting future direction.

## 5 SYSTEMATIC EVALUATION OF LLMS

Armed with BOOOOKSCORE, we now investigate the impact of several critical implementation decisions on summary coherence, including the choice of prompting strategy, base LLM, and chunk size. Overall, Claude 2 produces the most coherent summaries as measured by BOOOOKSCORE, followed closely by GPT-4 and distantly by GPT-3.5-Turbo, Mixtral-8x7B, and LLaMA2-7B-Inst; however, GPT-4's summaries are significantly longer and more detailed than the others across both prompting strategies. The rest of this section drills down into finer-grained results.

**Experimental setup:** Table 2 contains results for five instruction-tuned LLMs: GPT-4, GPT-3.5-Turbo, Claude 2, Mixtral-8x7B, and LLaMA2-7B-Instruct.[13] Unless otherwise specified, we set the chunk size to 2048, maximum summary length $G_n$ to 900, decoding temperature to 0.5,[14] and $p = 1$ for ancestral sampling.[15] To avoid confounds, we use identical prompts for all models except LLaMA2-7B-Inst, which only functions with a simpler prompt. LLM API costs for our experiments were $10K USD (Table 8); more experimental details are in Appendix D.

**Incremental summaries are almost always less coherent than their hierarchical counterparts.** Hierarchical summaries generally have higher BOOOOKSCORE than incremental summaries, likely because the incremental updating task requires the base LLMs to follow more complex instructions

---

[13] GPT-4 configurations in this table are not comparable to the ones we analyzed in Section 3 since we had to reduce chunk size and summary length due to LLaMA2-7B-Inst and GPT-3.5-Turbo's smaller context size.

[14] Claude 2 is the only exception, as we use its default temperature of 1.

[15] We use a temperature of 1 for compression, which improves adherence to the max summary length.

(e.g., deciding what to include from the current book chunk, what to discard from the summary, whether to restructure the summary, etc.). While hierarchical summarization potentially drops long-range dependencies, its instructions are generally simpler (summarize or merge).

**Incremental summarization benefits from increased chunk size.** The one exception to the above result is Claude 2 with a chunk size of 88K, whose incremental configuration produces slightly more coherent summaries than the hierarchical version (90.9 vs. 90.3 BOOOOKSCORE). In contrast, using Claude 2 for incremental summarization with a chunk size of 2048 results in a BOOOOKSCORE of 78.6, so clearly the model benefits from fewer updating and compression steps. We do not observe similar behavior with hierarchical summaries, which suggests that hierarchical book-length summarization is preferred for smaller context models.

**LLaMA 2 struggles on book-length summarization while Mixtral shows promising performance.** Table 2 shows that LLaMA-2-7B-Instruct achieves by far the worst hierarchical BOOOOKSCORE of any model. Its summaries also contain significant repetition (*% of repeated trigrams*), which is a critical coherence error. Furthermore, we could not get the LLaMA-2-7B-Instruct checkpoint to perform incremental updating at all, as it just copied text from the chunks until it reached the summary length limit, at which point it failed to follow the compression instruction. On the positive side, Mixtral-8x7B, another open-source LLM, outperforms LLaMA-2-7B-Instruct by a substantial margin, though it still trails behind most of the closed-source models. Nonetheless, it is encouraging to note that with performances closely matching that of GPT-3.5-Turbo on both summarization approaches, Mixtral-8x7B signals the narrowing gap between open-source and closed-source models.

**High coherence does not necessarily correlate with human preferences.** How well do coherence measurements from BOOOOKSCORE correlate with coarse-grained human preferences? We conduct another human evaluation study with the same four annotators in which we solicit preference judgments on pairs of GPT-4 generated incremental and hierarchical summaries.[16] As shown in Table 4, incremental summaries are almost always preferred over hierarchical summaries in terms of level of detail (83% vs. 11%). However, hierarchical summaries are preferred for better structure (59% vs. 35%), logical consistency (53% vs 38%), and overall (54% vs. 44%). When forming their overall preference, some annotators preferred the higher level of detail of incremental summaries at the expense of coherence; thus, both strategies can be viable depending on the needs of the user.

**Qualitative analysis:** Appendix L contains summaries generated from Janika Oz's *A History of Burning*, which tells a multi-generational story about an Indian family living in Uganda. We observe that both GPT-4 and GPT-3.5-Turbo tend to generate oft-repetitive and vague sentences within their summaries (e.g., *The story highlights the resilience and determination of the characters as they navigate the complexities of life, love, and identity across generations and continents.*). Such artifacts are rarely produced by the 88K chunk size version of Claude 2, which instead omits key information present in the beginning or middle of the input (e.g., the entire story of the first generation in the book) in favor of focusing on the end of the book, following the findings of Liu et al. (2023a). All configurations make faithfulness errors: for example, in *A History of Burning*, the mother of the character Hari is incorrectly identified as Rajni by Claude 2, while GPT-4 does describe Hari's parentage correctly at one point in the summary but incorrectly at another. We show in Appendix I that automatic quality metrics such as BLANC (Vasilyev et al., 2020) and SUPERT (Gao et al., 2020) are inadequate for book-length summarization.

## 6 LIMITATIONS

**Our error taxonomy is derived just from errors made by GPT-4.** We decided to conduct our human evaluations in Section 3 on summaries produced by GPT-4 for two reasons: (1) we wanted our error taxonomy to focus on errors that are actually made by state-of-the-art LLMs (unlike e.g., fluency errors present in SNaC); and (2) human evaluation is very costly, so we could not evaluate many different LLMs on our annotation budget. Similarly, we implement BOOOOKSCORE using GPT-4 as a base LLM, which may have some systematic biases that could be alleviated by using a pool of LLM annotators as in AlpacaEval (Dubois et al., 2023).

---

[16]Each annotator compared 25 disjoint pairs of summaries, and we paid $15 per task for a total of $1.5K. To prevent bias, we shuffle the ordering of incremental and hierarchical summaries for each summary pair, and conceal the summarization method of each summary.

**BₒₒₒₒₖSᴄᴏʀᴇ can be expensive to run.** Since computing BₒₒₒₒₖSᴄᴏʀᴇ requires iterating through a summary sentence by sentence using GPT-4, it can be expensive and slow especially given that the annotation prompt is long (see Appendix M.4). We did experiment with an approach that asked GPT-4 to annotate errors in the entire summary at once, but the generated annotations would often include too many trivial questions, and alignment with human judgments was low. That said, despite the API costs of GPT-4 and the relatively slow time to evaluate one summary, BₒₒₒₒₖSᴄᴏʀᴇ is still significant cheaper and faster than performing human evaluations.

**BₒₒₒₒₖSᴄᴏʀᴇ does not account for the relative importance of different error types.** Unlike similar evaluation frameworks such as MQM (Freitag et al., 2021), we choose not to assign severity weights to different error types. Nowadays, powerful LLMs rarely make errors related to grammar, which can be objectively defined. For other error types like those in our taxonomy, the notion of assigning relative importance is ill-defined. Furthermore, prior work (Goyal et al., 2022a; Dou et al., 2022) shows low recall between human annotations for NLG evaluation, which indicates that error type severity is subjective as annotators often do not highlight issues that others may find critical.

**No validation of recall.** Due to the expense, we do not collect overlapping annotations for each summary during human evaluation. Since the annotation task involves subjectivity, overlapping annotations can help ensure that all errors within a summary can be captured. However, recent work (Krishna et al., 2023) shows that a comprehensive annotation of all information units is not required to produce a useful aggregate score that can be used to rank different models.

## 7 Rᴇʟᴀᴛᴇᴅ ᴡᴏʀᴋ

**Book-length narrative summarization:** Most prior long-form summarization work still focuses on documents shorter than 10K tokens (Cohan et al., 2018; Kornilova & Eidelman, 2019; Wang et al., 2022). BookSum (Kryscinski et al., 2022) is the first published summarization dataset that includes book-level source text as part of their data, which encouraged modeling efforts in this direction (Wu et al., 2021; Xiong et al., 2022; Pang et al., 2023; Cao & Wang, 2023; Pu et al., 2023a).

**Fine-grained evaluation of generated text:** Our work relates to evaluation protocols within machine translation that annotate spans, error types, and error severities (Freitag et al., 2021; Fernandes et al., 2023), which are more meaningful than output ranking and Likert ratings. Also related is ACU (Liu et al., 2023c), an annotation protocol for summary salience evaluation that breaks summaries down into fine-grained content units, FactScore (Min et al., 2023), which dissects machine-generated text into atomic facts before evaluating their factual consistency, LongEval (Krishna et al., 2023), which includes an in-depth analysis of best practices for faithfulness evaluation in long-form summarization coherence evaluation, and SNaC (Goyal et al., 2022a), a coherence error taxonomy built for fine-tuned summarization models.

**Automatic evaluation with LLMs:** LLM evaluators have recently emerged as a cost-effective alternative to human evaluations, explored for both general conversational and instruction following capabilities (Dubois et al., 2023; Zheng et al., 2023) and traditional NLG tasks like summarization (Fu et al., 2023; Liu et al., 2023b; Wang et al., 2023). These latter studies substantiate LLMs' potential as an NLG metric, but only for evaluating short input-output pairs. In our work, we use GPT-4 to evaluate book-length summaries, uniquely employing a fine-grained automatic evaluation schema to set our work apart from existing research.

## 8 Cᴏɴᴄʟᴜsɪᴏɴ

Our work presents the first systematic study of book-length summarization using LLMs. We establish a novel human evaluation protocol to assess summary coherence on newly-published books. Then, we develop an LLM-based automatic metric called BₒₒₒₒₖSᴄᴏʀᴇ that relies on a coherence error taxonomy derived from our human annotations. Using BₒₒₒₒₖSᴄᴏʀᴇ allows us to evaluate various prompting strategies and model choices, revealing insights such as: hierarchical merging produces more coherent summaries but may lack detail compared to incremental updating; and increasing chunk size can significantly improve incremental updating. Interesting future directions include automatically evaluating faithfulness in the book-length summarization setting, benchmarking newer long-context LLMs using BₒₒₒₒₖSᴄᴏʀᴇ, and expanding BₒₒₒₒₖSᴄᴏʀᴇ to multilingual texts. We release our BₒₒₒₒₖSᴄᴏʀᴇ metric and annotated summaries to enable meaningful progress in book-length summarization.

## 9 ACKNOWLEDGMENTS

We extend special gratitude to members from the UMass NLP lab for participating in the pilot study and offering valuable feedback, and to the Upwork annotators for their hard work. This project was partially supported by awards IIS-2202506 and IIS-2046248 from the National Science Foundation (NSF) as well as an award from Open Philanthropy. We also thank the NSF's CloudBank program for supporting the majority of our LLM API-based experiments.

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

## A  DETAILS ON THE TWO PROMPTING STRATEGIES

Assume an LLM with context window size $W$ is used to summarize an input document $D$ whose length $L \gg W$. We thus split $D$ into non-overlapping chunks $c_1, c_2, \ldots c_{\lceil \frac{L}{C} \rceil}$ where $C < W$ is the length of each chunk.

### A.1  HIERARCHICAL MERGING

Hierarchical merging works as follows:

1. Obtain summaries at the base level $l = 0$ by summarizing each chunk.

2. Obtain summaries for the first level $l = 1$ by prompting the LLM to merge as many consecutive level-0 summaries $s_i, s_{i+1}, \ldots$ as possible[17] such that the total length of the merging prompt, the selected summaries, and the prior context (if there exists a preceding summary at the same level) is less than $W - G_l$, where $G_l$ is a hyperparameter controlling summary length that varies depending on the level $l$.

3. Repeat the previous step recursively until we are left with a single summary for the book.

### A.2  INCREMENTAL UPDATING

Incremental updating works as follows:

1. Feed the summarization prompt into the LLM along with the first chunk $c_1$ to obtain a summary of the first chunk, which initializes the global summary $g_1$

2. Now, provide the LLM with the updating prompt, the next chunk $c_2$, and the current global summary $g_1$. The model is prompted to updating the global summary to $g_2$ with information from the current chunk.

3. Iterate through the remaining chunks. If $g_i$ exceeds the maximum summary length $G_n$, call the compression prompt to compress $g_i$ to fit within the length limit.[18] See Appendix A.2.1 for more details on compression.

### A.2.1  COMPRESSION

The compression step is required for incremental updating. Through our experimentation, we have observed that as the model processes a book through incremental updating, it consistently adds more information to the running summary instead of removing things. Even with an updating prompt, the summary often surpasses the target length as removing content from it is not in the model's natural inclination. Thus, a separate prompt is needed for the model to condense the summary. However, in hierarchical summarization, condensing is not required. The merging step is less likely to run over the summary limit since it does not have to work with a pre-existing running summary. If the summaries generated during hierarchical merging go over the summary limit, simply asking the model to regenerate up to a fixed number of times would suffice.

## B  DATASET DETAILS

### B.1  TABLE OF ALL BOOKS IN THE DATASET

See Table 3.

---

[17]Wu et al. (2021) suggest that since independent chunk-level summarization might miss vital context from earlier sections of the story, we can mitigate this effect by joining as many preceding summaries from the same level as possible. We thus implement this approach in our method.

[18]As the final summary often contains artifacts like "in the current segment" or "in this section", especially with weaker models, we included an additional post-processing prompt to clean up these artifacts. See Appendix M.3.

Table 3: Title, author, genres, and publication date of all books in our dataset, sorted alphabetically.

| Name | Author | Genres | Published |
|---|---|---|---|
| A Day of Fallen Night | Samantha Shannon | fantasy, fiction, lgbt | 2023/02/28 |
| A Fever in the Heartland: The Ku Klux Klan's Plot to Take Over America, and the Woman Who Stopped Them | Timothy Egan | crime, historical, non-fiction | 2023/04/04 |
| A Girl Called Samson | Amy Harmon | fiction, historical, romance | 2023/04/01 |
| A Heart That Works | Rob Delaney | contemporary, family, memoir, non-fiction | 2022/11/29 |
| A History of Burning | Janika Oza | fiction, historical | 2023/05/02 |
| A House with Good Bones | T. Kingfisher | fantasy, fiction, gothic, horror | 2023/03/28 |
| A Likely Story | Leigh McMullan Abramson | contemporary, fiction, mystery | 2023/03/14 |
| A Living Remedy: A Memoir | Nicole Chung | biography, contemporary, memoir, non-fiction | 2023/04/04 |
| Age of Vice | Deepti Kapoor | crime, fiction, thriller | 2023/01/03 |
| All the Dangerous Things | Stacy Willingham | fiction, thriller | 2023/01/10 |
| Ander & Santi Were Here | Jonny Garza Villa | contemporary, fiction, lgbt, romance | 2023/05/02 |
| Atalanta | Jennifer Saint | fantasy, historical, mythology, retelling | 2023/04/11 |
| Black Cake | Charmaine Wilkerson | contemporary, fiction, historical, mystery | 2022/02/01 |
| Camp Zero | Michelle Min Sterling | fantasy, fiction, scifi | 2023/04/04 |
| Catfish Rolling | Clara Kumagai | fantasy, fiction, lgbt, mythology, scifi | 2023/03/02 |
| Central Places | Delia Cai | contemporary, fiction | 2023/01/31 |
| Chain-Gang All-Stars | Nana Kwame Adjei-Brenyah | fantasy, fiction, lgbt, scifi | 2023/05/02 |
| City Under One Roof | Iris Yamashita | crime, fiction, mystery, thriller | 2023/01/10 |
| Clytemnestra | Costanza Casati | fantasy, fiction, historical, mythology | 2023/05/02 |
| Deep as the Sky, Red as the Sea | Rita Chang-Eppig | fantasy, fiction, historical | 2023/05/30 |
| Did You Hear About Kitty Karr? | Crystal Smith Paul | fiction, historical | 2023/05/02 |
| Divine Rivals | Rebecca Ross | fantasy, historical, romance | 2023/04/04 |
| Drowning | T. J. Newman | fiction, mystery, thriller | 2023/05/30 |
| Emily Wilde's Encyclopaedia of Faeries | Heather Fawcett | fantasy, fiction, historical, romance | 2023/01/10 |
| Flowerheart | Catherine Bakewell | fantasy, fiction, romance | 2023/03/14 |
| Ghost Music | An Yu | contemporary, fantasy, fiction | 2022/01/01 |
| Good Night, Irene | Luis Alberto Urrea | fiction, historical | 2023/05/30 |
| Greek Lessons | Han Kang | contemporary, fiction, romance | 2023/04/18 |
| Greymist Fair | Francesca Zappia | fantasy, horror, mystery, retelling | 2023/03/28 |
| Gwen and Art Are Not in Love | Lex Croucher | fantasy, fiction, historical, lgbt, romance | 2023/05/11 |
| Happy Place | Emily Henry | contemporary, fiction, romance | 2023/04/25 |
| Heart of the Sun Warrior | Sue Lynn Tan | fantasy, fiction, mythology, retelling | 2022/11/10 |
| Homecoming | Kate Morton | fiction, historical, mystery, thriller | 2023/04/13 |
| Honeybees and Distant Thunder | Riku Onda | contemporary, fiction | 2023/05/02 |
| How to Turn into a Bird | María José Ferrada | contemporary, fiction | 2022/12/06 |
| I Have Some Questions for You | Rebecca Makkai | contemporary, fiction, mystery, thriller | 2023/02/21 |
| I Want to Die But I Want to Eat Tteokpokki | Baek Sehee | memoir, non-fiction | 2022/11/01 |
| In the Lives of Puppets | T. J. Klune | fantasy, fiction, lgbt, romance, scifi | 2023/04/25 |
| Into the Light | Mark Oshiro | contemporary, fiction, lgbt, mystery, thriller | 2023/03/28 |
| Isha, Unscripted | Sajni Patel | contemporary, fiction, romance | 2023/02/14 |
| Jana Goes Wild | Farah Heron | comedy, contemporary, fiction, romance | 2023/05/02 |
| Lady Tan's Circle of Women | Lisa See | fiction, historical | 2023/06/06 |
| Lies We Sing to the Sea | Sarah Underwood | fantasy, lgbt, mythology, retelling | 2023/03/07 |
| Lone Women | Victor LaValle | fantasy, fiction, historical, horror, mystery | 2023/03/28 |
| Lunar Love | Lauren Kung Jessen | contemporary, fiction, romance | 2023/01/10 |
| Maame | Jessica George | contemporary, family, fiction | 2023/01/31 |
| Meet Me at the Lake | Carley Fortune | contemporary, fiction, romance | 2023/05/02 |
| Natural Beauty | Ling Ling Huang | contemporary, fiction, horror, lgbt | 2023/04/04 |
| Paper Names | Susie Luo | contemporary, fiction, historical | 2023/05/02 |
| Pathogenesis: A History of the World in Eight Plagues | Jonathan Kennedy | historical, non-fiction | 2023/04/18 |
| Scattered All Over the Earth | Yoko Tawada | dystopian, fiction, scifi | 2022/03/01 |
| Secretly Yours | Tessa Bailey | contemporary, romance | 2023/02/07 |
| Seven Faceless Saints | M. K. Lobb | fantasy, fiction, lgbt, mystery, romance | 2023/02/07 |
| She Is a Haunting | Trang Thanh Tran | fantasy, fiction, gothic, horror, lgbt | 2023/02/28 |
| Some Desperate Glory | Emily Tesh | fantasy, fiction, lgbt, scifi | 2023/04/11 |
| Song of Silver, Flame Like Night | Amélie Wen Zhao | fantasy, fiction, mythology, romance | 2023/01/03 |
| Spare | Prince Harry | biography, memoir | 2023/01/10 |
| Spice Road | Maiya Ibrahim | fantasy, fiction, mythology, romance | 2023/01/24 |
| The Adventures of Amina al-Sirafi | Shannon Chakraborty | adventure, fantasy, fiction, historical | 2023/02/28 |
| The Bandit Queens | Parini Shroff | contemporary, fiction, mystery, thriller | 2023/01/03 |
| The Book of Everlasting Things | Aanchal Malhotra | fiction, historical, romance | 2022/12/27 |
| The Collected Regrets of Clover | Mikki Brammer | contemporary, fiction, romance | 2023/05/02 |
| The Covenant of Water | Abraham Verghese | fiction, historical | 2023/05/02 |
| The Faraway World | Patricia Engel | fiction, short stories | 2023/01/24 |
| The Ferryman | Justin Cronin | fantasy, fiction, scifi, thriller | 2023/05/02 |
| The First Bright Thing | J. R. Dawson | fantasy, fiction, historical, lgbt | 2023/06/13 |
| The Girl Who Fell Beneath the Sea | Axie Oh | fantasy, fiction, mythology, retelling, romance | 2022/02/22 |
| The Golden Doves | Martha Hall Kelly | fiction, historical | 2023/04/18 |
| The Half Moon | Mary Beth Keane | contemporary, fiction | 2023/05/02 |
| The Haunting of Alejandra | V. Castro | fantasy, fiction, historical, horror, mythology, retelling | 2023/04/18 |
| The House Is on Fire | Rachel Beanland | fiction, historical, mystery | 2023/04/04 |
| The Last Pomegranate Tree | Bachtyar Ali | fantasy, fiction, historical | 2023/01/03 |
| The Last Tale of the Flower Bride | Roshani Chokshi | fantasy, fiction, gothic | 2023/02/14 |
| The Marriage Portrait | Maggie O'Farrell | fiction, historical | 2022/09/06 |
| The Mimicking of Known Successes | Malka Older | fiction, mystery, scifi | 2023/03/07 |
| The Night Travelers | Armando Lucas Correa | fiction, historical | 2023/01/10 |
| The Stolen Heir | Holly Black | fantasy, romance | 2023/01/03 |
| The Survivalists | Kashana Cauley | comedy, contemporary, fiction, life, romance | 2023/01/10 |
| The True Love Experiment | Christina Lauren (duo) | contemporary, romance | 2023/05/16 |
| The Vibrant Years | Sonali Dev | contemporary, fiction, romance | 2022/12/01 |
| The Wager: A Tale of Shipwreck, Mutiny and Murder | David Grann | adventure, crime, historical, mystery, non-fiction | 2023/04/18 |
| The Wicked Bargain | Gabe Cole Novoa | fantasy, fiction, historical, lgbt | 2023/02/28 |
| The Wishing Game | Meg Shaffer | contemporary, fantasy, fiction, mystery, romance | 2023/05/30 |
| The Words That Remain | Stênio Gardel | contemporary, fiction, lgbt, romance | 2023/01/16 |
| The Writing Retreat | Julia Bartz | fiction, horror, mystery, thriller | 2023/02/21 |
| Things I Wish I Told My Mother | Susan Patterson, Susan DiLallo, James Patterson | contemporary, family, fiction, romance | 2023/04/10 |
| Thorne Princess | L. J. Shen | contemporary, fiction, romance | 2023/01/04 |
| Tomorrow, and Tomorrow, and Tomorrow | Gabrielle Zevin | fiction, gaming, romance | 2022/07/05 |
| Tress of the Emerald Sea | Brandon Sanderson | adventure, fantasy, fiction, romance | 2023/01/03 |
| Unseelie | Ivelisse Housman | fantasy, fiction | 2023/01/03 |
| Untethered Sky | Fonda Lee | fantasy, fiction, scifi | 2023/04/11 |
| Vera Wong's Unsolicited Advice for Murderers | Jesse Q. Sutanto | contemporary, fiction, mystery, thriller | 2023/03/14 |
| Victory City | Salman Rushdie | fantasy, fiction, historical, mythology | 2023/02/07 |
| Walking Practice | Dolki Min | fantasy, fiction, horror, lgbt, scifi | 2023/03/14 |
| We Don't Swim Here | Vincent Tirado | fiction, horror, lgbt, mystery, thriller | 2023/05/02 |
| What Happens Next | Christina Suzann Nelson | contemporary, family, fiction, mystery | 2023/01/17 |
| While Time Remains: A North Korean Defector's Search for Freedom in America | Yeonmi Park | biography, historical, memoir, non-fiction | 2023/02/14 |
| Witch King | Martha Wells | fantasy, fiction, scifi | 2023/05/30 |
| Wrong Place Wrong Time | Gillian McAllister | fiction, mystery, thriller | 2022/05/12 |
| Yellowface | R. F. Kuang | contemporary, fiction, mystery, thriller | 2023/05/16 |

# C    Coarse-grained human evaluation

Table 4 shows detailed results of the coarse-grained human evaluation as discussed in Section 5.

Table 4: Results from coarse-grained human evaluation. Annotators compare 100 pairs of GPT-4 generated incremental and hierarchical summaries and judge (1) overall preference; (2) level of detail; (3) structure and pacing; (4) logic and understandability of each summary.

| Preference | Incremental | Hierarchical | Tie |
|---|---|---|---|
| Overall | 44 | 54 | 2 |
| Detail | 83 | 11 | 6 |
| Structure | 35 | 59 | 6 |
| Logic | 38 | 53 | 9 |

## D  MORE EXPERIMENT DETAILS

We use the LLaMA-2-7B-Instruct checkpoint[19] fine-tuned on long-context summarization (Book-Sum). For Mixtral, we used the `mistralai/Mixtral-8x7B-Instruct-v0.1` checkpoint hosted by the Together API. For closed-source models, we use the `gpt-4 2023-03-15` and `gpt-3.5-turbo-0301` checkpoints on Microsoft Azure. Anthropic unfortunately does not disclose checkpoint information, but our summaries were all obtained via their Claude 2 API in September 2023.

In our experiments, LLaMA 2 uses a context window size of 4096 tokens, while other models leverage their full context window. While generating LLaMA 2 hierarchical summaries, we have to truncate the results at the final punctuation mark. If not, the model would be stuck at the regeneration phase, as it does not follow the given word limit at all. In addition, for LLaMA 2 summaries, we do not apply post-processing as described in Appendix D.1, because it would significantly alter the structure of the LLaMA 2 summaries, thereby enhancing their coherence. Instead, we post-process the summaries using a standard string matching approach to get rid of sentences copied from the prompt. Without this step, we cannot evaluate them with GPT-4, since GPT-4 would treat these prompt artifacts as instructions rather than sentences to annotate.

### D.1  DATA PROCESSING DETAILS

In order to preserve all visual separators within the text, we extract all text elements from the epub files without further automatic processing. As a result, sometimes content from non-narrative sections would appear in the generated summaries. We apply simple post-processing by prompting GPT-4 to remove information coming from non-narrative sections of the book. The prompt can be found in Appendix M.3.

## E  EXPERIMENTS WITH SQUALITY

To investigate the effect of incremental updating and hierarchical merging on summary coherence, we evaluated GPT-4 on the validation set of the SQuALITY dataset, which contains sci-fi stories that are 4000-6000 words long. Table 5 shows the ROUGE-L scores. The baseline setting is where the model summarizes the stories in one go, truncating the stories whenever they exceed the model's context window. Using incremental updating lowers ROUGE-L by a small amount, indicating that baseline summaries have slightly more overlap with the provided reference summaries. We will update the BOOOOKSCORE of these summaries in the next version of the paper.

---

[19]`https://github.com/togethercomputer/Llama-2-7B-32K-Instruct`

Table 5: ROUGE-L of summaries generated by GPT-4 on the SQuALITY validation set under two settings: baseline and incremental updating.

| Chunk size | ROUGE-L |
|---|---|
| *Baseline* | |
| - | 14.9 |
| *Incremental updating* | |
| 256 | 13.2 |
| 512 | 13.7 |

### E.1 QUALITATIVE ANALYSIS

We show the baseline, incremental, and human summaries for *Venus Is a Man's World* from SQuAL-ITY in Table 6. The incremental summary was generated by GPT-4 with a chunk size of 512.

The baseline summary provides the most coherent and comprehensive overview of the story's key events and themes. It clearly introduces the main characters - Ford, Evelyn, and Butt - and their backgrounds, while succinctly summarizing the plot including Ford's friendship with Butt, Evelyn's discovery and confrontation, and Butt's unconventional marriage proposal. The summary highlights central conflicts related to gender roles and norms. In contrast, the incremental summary, while touching on similar plot points and themes, does so in a more disjointed, less cohesive manner. Details about the setting, dunging drug harvesting, and weaponry feel redundant. Finally, the human summary lacks clarity in many parts, with confusing references to the Male Desuffrage Act and inconsistent character details. It focuses heavily on the early parts of the plot at the expense of later key events.

Overall, our finding that GPT-4 summaries can be better than human summaries aligns with results from recent work on LLM-based short-document summarization (Pu et al., 2023b). The qualitative analysis shows that methods like incremental updating could introduce coherence errors into LLM-generated summaries.

## F EFFECT OF SUMMARY LENGTH ON BOOOOKSCORE

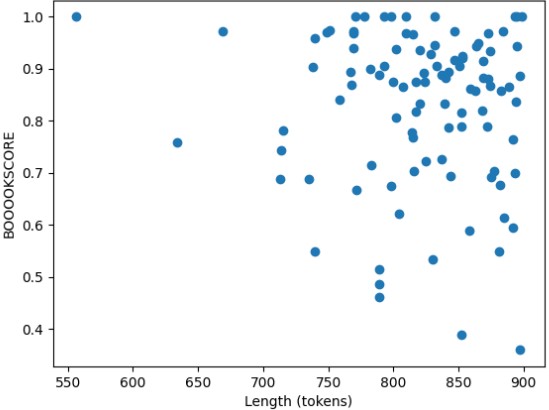

Figure 3: BOOOOKSCORE vs. summary length.

Table 6: Summaries of William Tenn's *Venus Is a Man's World*.

| Baseline | Incremental updating | Human |
|---|---|---|
| Venus Is a Man's World is a story about a young boy named Ferdinand, or "Ford," and his sister Evelyn, who are traveling on a spaceship called the Eleanor Roosevelt to the planet Venus. Evelyn is seeking a husband on Venus, as there is a shortage of men on Earth due to wars and other factors. Ford, curious and adventurous, explores the ship and stumbles upon a stowaway named Butt Lee Brown, a Venusian farmer who had gotten into trouble on Earth and is now hiding in a lifeboat.

Ford and Butt form a bond, with Butt teaching Ford about Venusian culture, songs, and even politics. Ford starts to admire Butt's strong, independent nature and begins to question the Earth's female-dominated society. However, when Evelyn discovers Ford's friendship with Butt, she confronts the stowaway, accusing him of corrupting her brother's morals and breaking the law by stowing away on the ship.

Butt, unfazed by Evelyn's accusations, proposes marriage to her, stating that they are both on the same journey to find a partner and that she has the qualities he is looking for in a wife. Ford excitedly urges his sister to accept Butt's proposal, but the story leaves the outcome of this unconventional proposal unresolved. The story highlights the clash of gender roles, societal norms, and the challenges faced by individuals seeking love and companionship in a rapidly changing world. | In the story, siblings Ford and Evelyn from an underwater city accompany women seeking husbands on a journey to Venus. Ford befriends stowaway, Butt Lee Brown, who has a troubled past. They bond over their life stories as Butt educates Ford about Venusian culture and weaponry. Despite being outsiders, they travel together.

Evelyn convinces Ford to attend a geography lecture, where she becomes deeply engaged in the discussion of illegal dunging drug harvesting on the Macro Continent. Ford's knowledge of dunging and the Venusian term "Paddlefoot" piques Evelyn's interest. Ford argues that Venusians are pioneers who need strong, independent men for building a civilization on the new planet, thus intriguing Evelyn.

Ford takes Evelyn to meet Butt Lee Brown, making her promise not to turn him in. Upon meeting, Evelyn confronts Butt about his illegal activities and moral delinquency. Butt dismisses her concerns, sparking a debate between them about Earth and Venusian societies. During their conversation, Butt proposes to Evelyn, stating that they both have similar goals in finding a spouse and that she possesses qualities he finds desirable. Evelyn is taken aback by the proposal, and her brother Ford excitedly encourages her to accept.

The story continues to explore the characters' relationships and their differing perspectives on the new planet, as well as the potential consequences of their actions and decisions. | Ferdinand is on a spaceship with his elder sister. The ship is jam-packed with females going to Venus in search of husband and he exits the cabin while the women are still in their hammocks. The women at his times have all the rights and do all the important things since the Male Desuffrage Act, so the boy is admitted to the ship on behalf of his sister. He explores the empty ship in search of portholes and after some hesitation enters a forbidden area. There he looks at the stars and then tries to open a lock on the lifeboat. A huge scary man appears with a blaster and frighteningly cold gaze. Ferdinand explains that he comes from Undersea, an area on Earth, and tells his family story - his parents being one of the first married couples in Undersea and dying a while ago, leading to his sister's decision to migrate to Venus. The stranger, Butt, tells about the lack of women on Venus and his travel to Earth in search of a wife without any idea "it's a woman's world". So he got in trouble with the law and stowed away on this ship. His many brothers were killed in a rising and only one is left. From that day on Ferdinand keeps visiting the stowaway bringing fruits and listening to stories about Venus. Butt teaches the boy to use the blaster without giving it not hold and constantly asks about Evelyn, the sister. Once, Ferdinand attends a geography lecture on the ship with his sister and corrects the lecturer about Venusian geography. Evelyn starts eliciting where the boy learned that and the boy tells about real men working on Venus. Sis gets angry with those masculine ideas and doesn't believe them to come from a little boy. Ferdinand tries to lie but Sis suppresses him into confession and he leads her to Butt. She tells Butt about all the laws he has broken while the least responds with an appeal to sense. Suddenly, Butt simply proposes a mutually beneficial marriage to stop the debate. |

To explore the impact of summary length on BOOOOKSCORE, we plot BOOOOKSCORE vs. length in Figure 3 for the 100 summaries generated by GPT-4 with incremental updating using a chunk size of 2048. The plot indicates that there is no discernible correlation between length and BOOOOKSCORE.

## G   BOOTSTRAPPING ANALYSIS OF BOOOOKSCORE

To check the stability of the BOOOOKSCORE metric, we ran a bootstrapping experiment. Given the BOOOOKSCORE of 100 summaries generated by GPT-4 using incremental updating with a chunk size of 2048, we randomly sample 1000 times with replacement using a sample size of 100, then compute the mean of these 1000 samples. The standard deviation of these samples is 0.015, which suggests that BOOOOKSCORE is consistent and reliable across multiple random samples.

# H  LABEL-WISE ALIGNMENT BETWEEN GPT-4 AND HUMAN ANNOTATIONS

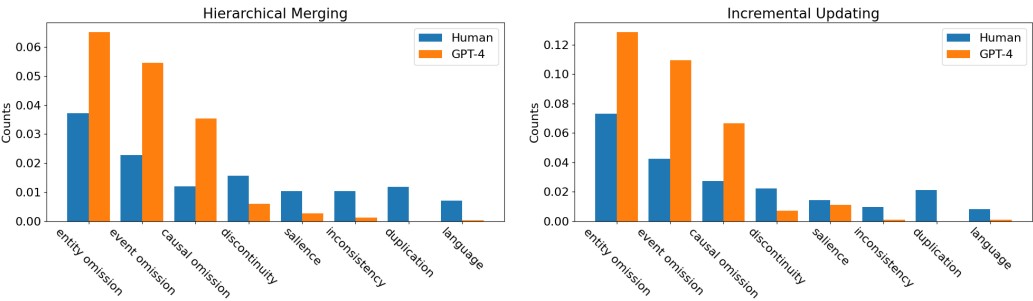

Figure 4: Distribution of all error types for incremental and hierarchical summaries generated by GPT-4, normalized by the total number of sentences in the summaries.

In our approach to GPT-4 automatic evaluation, we incorporate error type prediction into the prompt to assist the model in making reasoned judgments. We analyze the label-wise correlation between GPT-4 and human annotations, presenting the results in Figure 4. Despite the high agreement between precision as discussed in Section 4.1, GPT-4's error distributions vary significantly from those of human annotations. GPT-4 demonstrates a strong propensity for labeling omission errors, occasionally applying these labels to sentences that could be more appropriately categorized under different error types. In this study, we have not delved deeper into the error types predicted by GPT-4 given the observed inaccuracy. Further analysis of GPT-4's capability to precisely predict error types remains a potential area for future research.

# I  EFFECTIVENESS OF EXISTING REFERENCE-FREE EVALUATION METRICS

Table 7: Results from BLANC and SUPERT on all *hierarchical* summaries.

|  | BLANC | SUPERT |
|---|---|---|
| GPT-4 (4096) | 0.0248 | 0.3627 |
| GPT-4 (2048) | 0.0221 | 0.3615 |
| GPT-3.5-Turbo (2048) | 0.0203 | 0.3658 |
| Claude 2 (88000) | 0.0171 | 0.349 |
| Claude 2 (2048) | 0.0167 | 0.3559 |
| Mixtral-8x7B (2048) | 0.0169 | 0.3836 |
| LLaMA2-7B-Inst (2048) | 0.013 | 0.3446 |

We investigate how well existing reference-free evaluation metrics work for our setting where the source text is at book level. BLANC (Vasilyev et al., 2020) measures how helpful a summary is to understanding the source document by testing if a pre-trained model can better fill in masked words given access to the summary. SUPERT (Gao et al., 2020) measures the semantic similarity between the a summary and some pseudo reference summaries generated from the source document. We compute both metrics for hierarchical summaries generated under all configurations, and show results in Table 7. Both BLANC and SUPERT differ by nearly negligible margins across models, which makes them less meaningful. Furthermore, both metrics rate Claude 2 (88000) summaries as inferior to GPT-3.5-Turbo (2048) summaries, a finding that clearly contradicts results from our qualitative analysis.

We do not compute ROUGE (Lin, 2004) as it has been shown to have significantly low human correlation when applied to summaries generated by GPT-3 (Goyal et al., 2022b).

## J  COMPUTING ANNOTATION PRECISION

In this human evaluation task, given a summary and an annotation (span-question pair) for the summary, we ask annotators whether they agree, partially agree, or disagree with the annotation. We provide the following *standards* for determining agreement:

1. The span is confusing.

2. The questions are:

   (a) Relevant to the span.

   (b) Not answered anywhere in the summary, explicitly or implicitly.

   (c) Addressing issues that if left unresolved, would make the summary incoherent or make it hard for readers to understand the main storyline.

Agreement would mean both the span and the questions satisfy the *standards*. Disagreement would imply that the span is not at all confusing, regardless of the questions. Partial agreement would apply when the span is confusing, but the questions fail to meet one or more of the *standards*.

We mix human and GPT-4 annotations in the data we present to the annotators, concealing the origin of these annotations to prevent any potential bias. Each annotator was responsible for the human and GPT-4 annotations of 25 books (different from the 25 books which they annotated summaries for). We paid $1.7 per summary and $0.15 per annotation. 100 summaries and 1659 annotations resulted in a total cost of $418.85 (USD). Note that annotators were informed of this evaluation task weeks after they completed the first task (where they were asked to highlight spans and ask questions given book summaries). Thus, monetary reward for the second task could not have biased annotators to produce as many annotations as possible for the first task.

When computing precision, we count all cases of agreement and partial agreement.

## K  API COSTS

We present an estimation of the API costs of our experiments in Table 8.

Table 8: API cost estimates in USD.

| Model | Chunk Size | Summarization | BOOOOKSCORE Eval |
|---|---|---|---|
| *Summaries generated via hierarchical merging* | | | |
| GPT-4 | 4096 | 580.3 | 566 |
| GPT-4 | 2048 | 628.3 | 394.1 |
| GPT-3.5-Turbo | 2048 | 29.5 | 379.8 |
| Claude 2 | 2048 | 267.2 | 379.5 |
| Claude 2 | 88000 | 227 | 378.9 |
| Mixtral-8x7B | 2048 | - | 460.5 |
| LLaMA2-7B-Inst | 2048 | - | 538.5 |
| *Summaries generated via incremental updating* | | | |
| GPT-4 | 4096 | 786.7 | 797.5 |
| GPT-4 | 2048 | 873.2 | 536.8 |
| GPT-3.5-Turbo | 2048 | 69.7 | 318.4 |
| Claude 2 | 2048 | 329.6 | 356.6 |
| Claude 2 | 88000 | 229 | 518.7 |
| Mixtral-8x7B | 2048 | - | 395.7 |
| **Total** | | **4020.5** | **6021.2** |

# L  EXAMPLE SUMMARIES

Table 9: GPT-4 4096 summaries of Janika Oz's *A History of Burning*.

| Strategy | Summary |
|---|---|
| Hier | A History of Burning follows the lives of an Indian family living in Uganda during the late 19th and early 20th centuries, experiencing political turmoil, displacement, and personal struggles. The story begins with Pirbhai, a young boy from Gujarat, India, who is lured into indentured servitude and forced to build railways for the British in East Africa. He eventually marries Sonal, and they settle in Uganda, having a son named Vinod. |
| | Vinod grows up feeling the pressure to provide for his family and takes a job at a tea-coffee trading company. He marries Rajni, a young woman from Karachi, and they have two daughters named Latika and Mayuri. Latika grows up in Uganda, facing discrimination and violence during Idi Amin's reign, while Mayuri decides to study medicine in Bombay, India, leaving the family behind. As the political situation in Uganda worsens, the family is subjected to the "cattle count" census. Despite these hardships, the family remains resilient and determined to survive. |
| | Latika becomes involved in political activism, writing for a subversive newspaper called Jicho, and falls in love with Arun, a charismatic student activist who boards with her family. However, Arun becomes disillusioned with his work, and their relationship becomes strained. Latika enlists the help of Irene, an African woman who works for Arun's family, and her husband Kariuki to distribute her writings for Jicho. The two women form a bond based on their mutual struggle for justice and equality. |
| | Years later, Vinod, Rajni, and their family are living in Toronto, Canada, after being forced to leave Uganda under Idi Amin's rule. They struggle to adapt to their new lives, maintain connections to their homeland, and navigate the complexities of their relationships. Rajni works as a nanny for a woman named Jane and her daughter Dahlia, while Vinod works a low-paying job. Their daughter Kiya has become distant and withdrawn, while Mayuri is now married to Kunal, a man she barely knows. |
| | Mayuri returns to Toronto after years of living in Bombay, where she had studied medicine and formed a relationship with a woman named Ruhi. She has married Kunal to be able to return to her family. Upon arriving in Canada, Mayuri is struck by the differences in the weather and environment compared to India. She is also surprised by how much her family has changed over the years. Mayuri's relationship with her sister Kiya is strained, as Kiya is resentful of Mayuri's absence and her sudden return. |
| | One day, Kiya receives a letter from Adroa, her former lover, revealing that her long-lost sister Latika may still be alive in Uganda. The letter brings back memories and emotions for Kiya, who struggles to process the information. As she contemplates the news, her family life continues around her. Kiya decides to share the news with her father, who is devastated by the revelation. |
| | As the family tries to cope with the revelation, Vinod and Rajni decide to move into the apartment above the convenience store owned by their late friend Pulinbhai and his wife Nainabhen, leaving their house to their children. Hari, who has grown sullen and restless, is grappling with feelings of abandonment and anger over the family's secrets. Despite the turmoil within the family, they continue to support each other, with Rajni and Vinod trying to reach out to Hari and help him through his pain. |
| | In the following years, Hari becomes involved in a protest against racism, which takes a violent turn as the police attempt to disperse the crowd. Vinod's health deteriorates, but he finds solace in the thought of spending time with his family in their garden. Kiya and Mayuri receive a phone call informing them that Hari is missing and join their parents in searching for him. As they search, they reflect on their own experiences and the sacrifices their family has made for a better life. |
| | The story then shifts to an older woman and a young man standing by a lake, discussing their shared history and experiences. Both have dealt with separation and loss, and they find solace in each other's company. They speak of fires old and new, and the woman shares a tale from her past. The man recalls his own losses, and they stand together, acknowledging the cycle of life and the importance of knowing one's history. |
| | Throughout the story, the characters grapple with the challenges of living in a foreign land, maintaining their ambitions and careers, and navigating the complexities of their relationships with each other and the larger political events happening around them. Despite the obstacles they face, they remain determined to carve out a meaningful life for themselves and continue their fight for a better future. The novel explores themes of family, separation, and the importance of understanding one's history and lineage to achieve a greater understanding of personal identity. The family's history of burning and rebuilding serves as a metaphor for their resilience and determination to create a better future. |
| Inc | In 1898, Indian boy Pirbhai is tricked into working on a railway in Mombasa, Kenya. He finds shelter with Sonal's family in Kisumu and marries her, starting a family. Sonal is a gifted healer, and their son Vinod struggles to balance his Indian heritage with his friendships. Inspired by his uncle Nanu-mama, Vinod stays in Uganda, joining a tea-coffee trading company. |
| | Pirbhai arranges Vinod's marriage to Rajni in Karachi. Rajni is apprehensive about leaving her home, but her parents see it as an opportunity to escape the violence of Partition. Rajni marries Vinod and gives birth to a daughter. The couple faces challenges together and grow closer, ultimately finding strength in their growing family. When the third granddaughter is born, Nanu-mama visits, fleeing violence in Kenya. They support him despite the initial tension. As Pirbhai's life nears its end, he shares memories and wisdom with Latika, emphasizing resilience and their family's history. |
| | The family takes in Arun, a university student and paying guest. He grows closer to Latika, the oldest granddaughter. Arun offers her a tour of the campus, exciting the household. Latika joins him in a political rally which turns violent but remains active alongside Arun. They marry despite parental disapproval and move to Jinja, Uganda. Latika and Arun work to distribute her writings on unity and resistance. |
| | Mayuri, Latika's sister, is encouraged to study abroad and attends a Navratri celebration where she finds a connection with a girl who teaches her to dance with dandiyas, leading her to study medicine in Bombay. Her younger sister, Kiya, becomes close to Adroa, a young man working at an auto shop. The family goes to a camp for an Asians census but is forced to restart the line. Kiya feels lonely as Adroa appears distant and has an encounter with a harassing soldier. Adroa later reveals he has been asked to join Amin's army, causing a rift between them. |
| | At Mayuri's goodbye dinner, tensions arise between Kiya's family and Adroa. Kiya's mother calls her a "veshya," causing Kiya to break down. Adroa leaves, and Kiya finds solace in her sisters' embrace. Vinod encounters a soldier named Moses and is warned to get his kipande soon. Vinod is stopped by soldiers who force him and other men to lie on the ground, but a soldier recognizes him as a man of God and lets him go. |
| | Arun receives a refugee card, giving them seventy-two hours to leave Uganda. He discovers a letter from the government ordering the termination of Latika's newspaper, Jicho, but decides not to confront her about it. Their time in Uganda runs out, and Latika stays behind to search for Arun, who was taken away by armed men. She entrusts her child, Harilal, to Rajni, as they leave Uganda. |
| | The family adjusts to life in Toronto, coping with the cold weather and cultural differences. Rajni becomes friends with neighbor Jane, while Mayuri returns to Toronto to reunite with her family. Harilal is placed in a special class for gifted students, forming a friendship with a boy named Solomon. Kiya becomes concerned about their class and investigates the situation. |
| | Vinod becomes interested in buying a house, seeking stability. He visits an open house and is excited about the prospect of a new beginning. An accident occurs while Harilal is helping Vinod paint the house. Harilal falls off a ladder and is rushed to the hospital with a broken ankle. Vinod and Rajni reflect on their roles as parents and the importance of family and faith. |
| | Rajni receives a phone call from Kantabhen, Arun's mother, revealing that Latika is alive and living in London. Kantabhen wants to give back the property that her husband had transferred to Arun. Rajni and Vinod decide to tell Hari the truth about his real parents. Hari is angry and feels betrayed by their deceit. The family struggles to come to terms with the revelation and confronts the challenges it presents. |
| | Hari spends his days playing basketball with his friends, including Solomon, who is struggling with his job and family issues. Hari reflects on his family's past and the phone number that Ma had given him to contact Latika. As he and his friends continue to play basketball, they notice that the hoops have been removed from the court, symbolizing the loss of their sanctuary and the challenges they face. |
| | Hari considers the implications of reaching out to Latika and the sacrifices his family has made to give him a chance at a better life. He realizes the importance of family and the power of forgiveness, choosing to embrace the truth and the complexities of his relationships with his parents, sisters, and friends. As the family continues to navigate the challenges of their past, present, and future, they find strength in their connections and the resilience that has been passed down through generations. |
| | Hari and Solomon confront the loss of their basketball sanctuary and rally together, deciding that if they were to rise, it could only be together. Hari's relationship with his family grows stronger as he recognizes the thread of trust, love, and time that has held them together. Rajni's perspective on her family's history and struggles evolves, and she begins to understand the importance of bringing the past forward. The family stays strong, supporting one another through life's difficulties as they continue to navigate the balance between their heritage and their new lives in Toronto. |

Table 10: GPT-4 2048 summaries of Janika Oz's *A History of Burning*.

| Strategy | Summary |
|---|---|
| Hier | "A History of Burning" by Janika Oza follows an Indian family living in Uganda through multiple generations as they navigate hardships, sacrifices, and cultural identity struggles. The story begins with Pirbhai, a 13-year-old boy from Gujarat, India, who travels to Africa in search of work during a time of drought and hardship. He eventually marries Sonal, a young girl living with her family in Kenya, and they move to Kampala, Uganda, where they have a son named Vinod.

Vinod helps elevate his family's status in Uganda and marries Rajni, a 19-year-old girl from Karachi, amidst rising tensions between Hindus and Muslims. They raise two daughters, Latika and Mayuri, while facing challenges in their relationship and adapting to their new surroundings. Throughout the story, the characters face hardships, sacrifices, and the struggle to maintain connections to their homeland while navigating the challenges of colonialism, migration, and cultural identity.

Years later, Latika grows up and becomes involved in activism against colonial rule in Africa, writing for a secret network called Jicho. She marries Arun, a fellow activist, but struggles with her new life living with his family and the expectations placed upon her. Meanwhile, her sister Mayuri navigates a newly integrated school in Uganda, facing racism and discrimination while considering her future. Eventually, Mayuri studies medicine in India and later returns to her family in the U.S., accompanied by her partner Kunal.

The family faces challenges as political changes in Uganda, the Mau Mau rebellion in Kenya, and the lingering effects of British colonialism impact their lives. Latika secretly writes for an anti-government newspaper, JICHO, and her husband Arun is kidnapped due to her involvement. Mayuri is surprised by the changes in her family, particularly her parents' aging and her younger siblings, Kiya and Hari. She learns that Hari, the son of her deceased sister Latika and her husband Arun, is unaware of his true parentage.

As the family members struggle to maintain connections to their homeland and each other, they experience loss, love, and resilience in the face of adversity. Throughout the story, the characters find strength and hope in their shared experiences and love for one another, as they try to build a better life for themselves and their families. The story also explores the challenges faced by immigrants as they adapt to new environments, such as Mayuri's struggle to practice medicine in Canada and the family's experiences with racism and violence.

In the end, the family must come together to face the destruction of their shop and the loss of their past, finding solace in each other and the hope for a better future. Through their shared experiences and love for one another, they find strength and hope to endure the challenges that come their way. The story highlights the resilience and determination of the characters as they navigate the complexities of life, love, and identity across generations and continents. |
| Inc | Indian immigrant Pirbhai moves his family from Kenya to Uganda, where his son Vinod marries Rajni from Pakistan. They have three daughters, Latika, Mayuri, and Kiya, and face hardships under General Idi Amin's rule. Forced to leave Uganda, they settle in Toronto, where Vinod and Rajni face cultural challenges. Mayuri marries Kunal but struggles to adapt and returns to her family when her medical degree is unrecognized. Kiya becomes unexpectedly pregnant and faces shame and isolation. The story alternates between Kiya and Mayuri's perspectives in Toronto and Latika's life in London.

Vinod and Rajni buy a house in Toronto, providing stability for their family. They find a sense of belonging and hope but still worry about their daughters. Latika struggles in London, gaining insight into her past through a chance encounter with a woman from Uganda. Encouraged by her mother-in-law, Latika considers contacting her family. Back in Toronto, Rajni learns that their soap factory in Uganda may be returned to them under a repatriation scheme. She also discovers that Latika visited Uganda, shocking the family with the revelation that she is alive. The family confronts the secrets they've kept over the years.

Rajni tells her brother Harilal the truth about Latika and their parents' past as political activists. The family dynamic becomes strained as Harilal reacts with anger and shock. Vinod and Rajni move into an apartment above a friend's convenience store to give their children space. They find purpose and develop a routine, while Harilal's attempts to reconnect are met with resistance.

Mayuri, now living in her family's house, tries to maintain normalcy but struggles to connect with Harilal after revealing the truth about Latika. She befriends Darlene, a Trinidadian caretaker who shares her experiences of displacement and finding a new home. Through gardening and connecting with neighbors, they find solace and a sense of community.

As the family confronts their past, Kiya tries to hold them together and find forgiveness. They find healing and understanding by sharing stories, exploring themes of family bonds, secrets, and adaptation. Mayuri's son Hari struggles with the family's secrets and changes in their relationships. He finds solace in playing basketball with friends who understand his feelings of displacement.

Hari's basketball court is taken away, symbolizing his struggle for belonging and justice. His family tries to reconnect with him, and he begins to understand their silence as a means of survival. Hari learns about a protest against racial injustice and police brutality and decides to participate. The protest becomes chaotic, and Hari is separated from his friends and blinded by tear gas.

As the chaos unfolds, Rajni and Vinod search for Hari. The family reunites and witnesses their convenience store burning down, but they resolve to rebuild and support one another. They come to terms with their past and find strength in their shared experiences, such as gardening and connecting with their community.

In the epilogue, Latika and her husband reflect on their lives and the stories they've shared. They find solace in their connection and the knowledge that what the water takes, it returns. The family's journey is marked by resilience, forgiveness, and the power of community to heal and rebuild.

The novel also explores the importance of knowing one's history and family lineage to achieve a greater understanding of personal identity. Each family member faces moments where they must decide between complicity and resistance, and their family legacy influences their decisions. The novel emphasizes the power of storytelling and the interconnectedness of the characters as they navigate their relationships, secrets, and the challenges they face. Food plays a significant role in the narrative, symbolizing community, belonging, and the characters' connection to their cultural roots. At the heart of the novel is the idea of community and collectivism, as well as the question of who is included and excluded in these spaces. The author hopes readers will reflect on these themes and the possibilities of solidarity, as the characters' fates are entangled and they rely on one another. |

Table 11: GPT-3.5-Turbo 2048 summaries of Janika Oz's *A History of Burning*.

| Strategy | Summary |
|---|---|
| Hier | "A History of Burning" by Janika Oza is a multi-generational story that follows the lives of Pirbhai, Sonal, and their children Vinod, Rajni, Mayuri, Kiya, Meetu, and Hari through displacement, cultural struggles, and personal tragedies. The novel is set during political unrest in Kenya and Uganda in the mid-twentieth century, where the family faces poverty and exploitation as migrant workers. Pirbhai leaves his family in India to work in Kenya, where he faces poverty and exploitation as a migrant worker. He is haunted by the guilt of leaving his family behind and is forced to make difficult choices, including burning down a cluster of abandoned huts. Pirbhai marries Sonal, and they start a family, facing financial struggles and the impact of colonialism. |
| | The story explores themes of poverty, desperation, and the harsh realities of migrant workers. The family faces political unrest and racial tensions in Uganda during the time of the country's independence from British rule. They are forced to leave Uganda due to the political instability caused by Idi Amin's regime. The story also delves into Mayuri's struggles as a medical professional facing discrimination in Toronto. The characters are motivated by their desire to protect their families and survive in a dangerous environment, with themes of family, resilience, and the challenges faced by immigrants in adapting to new cultures. |
| | The story follows their children, Vinod and Rajni, and their struggles with identity, cultural expectations, and displacement. The family's relationships and dynamics are explored, and their motivations and fears are hinted at. The family lives in Uganda during the time of the country's independence from British rule, and the story explores themes of grief, family, and tradition. Pirbhai and Sonal celebrate the birth of their third granddaughter, Kiya, and plan a picnic at the Botanical Gardens in Entebbe. The family faces political unrest and racial tensions, with Pirbhai's granddaughter Latika becoming involved in a resistance movement against tyranny. The story features non-linear narratives and flashbacks, with Pirbhai reflecting on his past and the consequences of his actions. |
| | The novel shifts between different characters' experiences, flashbacks, and tragedies, providing a rich and layered narrative. Specifically, the characters' past traumas, including the disappearance of Latika, add complexity to their struggles and sense of loss, while moments of triumph and fear reveal the human capacity for resilience and transformation. Beyond the family's personal struggles, the novel delves into themes such as community inclusivity, belonging, and solidarity. |
| | Despite the family's struggles, the story presents moments of hope and triumph, such as Vinod's dream of buying a house, and moments of uncertainty and fear, such as Hari's fall from a ladder. Ultimately, the story presents a nuanced and emotionally resonant portrait of immigrant life and the human capacity for resilience and transformation. The characters experience moments of hope and triumph, such as Vinod's dream of buying a house, and moments of uncertainty and fear, such as Hari's fall from a ladder. |
| | The setting is in Uganda, Kenya, and Canada, and the story highlights the importance of family, resilience, and the challenges faced by immigrants in adapting to new cultures. The characters are motivated by their desire to protect their families and survive in a dangerous environment. The story presents a consistent and chronological narrative, despite featuring non-linear narratives, flashbacks, and switches between alternate worlds or viewpoints. |
| | In conclusion, "A History of Burning" is a complex and emotionally resonant story that explores themes of family, resilience, displacement, and the challenges faced by immigrants in adapting to new cultures. The story follows the lives of Pirbhai, Sonal, and their children through political unrest, racial tensions, and personal tragedies. Despite the family's struggles, the story presents moments of hope and triumph, ultimately presenting a nuanced and layered portrait of immigrant life and the human capacity for resilience and transformation. |
| Inc | A History of Burning follows the lives of nine family members over a century, exploring the importance of knowing one's history and family lineage to achieve a greater understanding of personal identity. The novel emphasizes the significance of community and belonging, while also considering who is included and excluded in these spaces. Food plays a pivotal role in placemaking and finding one's way into new understandings of home and belonging. The protagonist of the novel is the family itself, and the choice of whose voice to enter for specific moments was often intuitive. |
| | Vinod's health is declining, and he suffers a stroke. His family worries about him, and Rajni struggles with memories of the past, including the secret of her lost daughter, Latika. Mayuri and Kiya tell Hari and Meetu about their lost sister, and they all eat cake together. Hari confides in Sol about his boss accusing him of stealing tips, and they start playing basketball together. |
| | While Hari marches in a protest with Solomon, Marcus, and Sami, the protest becomes violent, and Hari is blinded by a searing fog. He stumbles on, trying to feel his way unseeing over whose streets. Hari yells out the names of his family, who have come and gone, and sees visions of a body huddled in a dank chamber, a tree blooming red as flame, and a boat dimly bobbing across black waters. The chemical fog hangs like a white sheet in the night sky, and Hari opens his eyes to see the destruction around him. He runs home, and his family is relieved to see him. |
| | Later, Vinod and Rajni meet with a woman on a rocky outcropping off the shore. They recognize each other and speak of people remote and unknowable as the land before the water. The woman has traveled far, and yet in all the movement, a part of her feels restored. Encircled by her family, she knows that she has come not just forward, but also back. When their conversation falters, they take in the stillness around them, how the waves make silence with sound. They let the tide crest over their feet and remember what the water has carried and what it has swallowed. The man slips a little on a rock slick with algae, and the woman rights him in a gesture unpracticed and natural. They recall losses that they cannot know. Beyond them, the tide continues. The water shifts the sand, reassembles the shells, sucks the stones to pebbles. The waves break and mend, break and mend. |

Table 12: Claude 2048 summaries of Janika Oz's *A History of Burning*.

| Strategy | Summary |
|---|---|
| Hier | The novel follows an Indian family across multiple generations, beginning in late 19th century India when the patriarch Pirbhai boards a ship to Africa seeking work. Tricked into indentured labor in Africa, Pirbhai endures difficult conditions building the Uganda Railway. Years later in 1902, his granddaughter Sonal notices the strong arms of an Indian laborer, Pirbhai, who comes asking for work at her father's shop in Nairobi. Sonal develops an interest in Pirbhai, who remains quiet about his traumatic past. |
| | The story shifts to 1917, where Sonal is now a married woman with two daughters, living in Kampala, Uganda. Her husband Pirbhai insists on sending money to his aging mother in India, while Sonal resents having another child amidst financial struggles. In the 1940s, their daughter-in-law Rajni secretly poisons her husband Vinod to make him too ill for military conscription during World War II, afraid he will die in combat. |
| | Years later, another descendant, Latika marries a local activist named Arun, against her parents' wishes. During political turmoil in the 1960s, Arun is imprisoned for sedition. Latika flees Uganda for London, abandoning their infant son. Decades later, Latika's sister Mayuri arranges her son Hari's marriage to a woman named Rajni, keeping secret that Rajni is actually Latika's daughter-in-law. In the 1970s, President Idi Amin expels Asians from Uganda, forcing the family to scatter across continents. They lose their home, bonds, and history in the upheaval. |
| | In 1980s Canada, Rajni and Vinod struggle to assimilate while hiding the truth about Hari's parentage. Hari feels disconnected from his family's silenced history. He grapples with fighting racism alongside friends versus fulfilling his duties to their shop and ill grandfather. When Hari finally learns about his mother Latika, he is furious over the lies. The family ruptures further when Latika mysteriously returns after 30 years, devastating Rajni. |
| | They continue navigating the legacy of Latika's unexplained absence and the cultural erasure caused by their exile from Uganda. But glimmers of reconciliation emerge, as when Rajni and Vinod open a new shop providing community. Later, while caring for their bedridden grandfather, Hari realizes his family's perseverance outweighs the hidden past. Ultimately, Latika's return signifies the possibility of mending fissures left by displacement and secrecy. Spanning a tumultuous century, the epic novel conveys a family's resilience and quest for home and belonging amidst upheaval. |
| Inc | Rajni cared for her ailing husband Vinod after his stroke. Their adopted son Hari moved back in to help, though Rajni still felt distant from him after revealing his birth mother Latika was alive. Rajni heard warnings about potential violence at an upcoming protest rally, but let Hari decide to attend and support his marginalized community. |
| | Rajni reflected on her family's complex history and relationships. She considered her daughters Kiya and Mayuri, who had tried in their own ways to heal family rifts. Rajni realized she had once been close with Hari and longed for that again. She thought of the connections between her family's struggles and those of other marginalized groups like Hari's friends. Rajni gained perspective on the importance of open communication to bring people together across divides. |
| | Meanwhile, Vinod slowly recovered, regaining mobility and strength. He woke up determined to become more independent. Vinod and Rajni both acknowledged their growing frailty, but continued supporting each other. |
| | Vinod's daughters Kiya and Mayuri tended a community garden, bringing together neighbors and friends. Though facing opposition, Kiya was inspired by her ancestor's vision of a shared place welcoming to all. She hoped the garden could be part of creating an inclusive home. |
| | Latika had told them the story of how their grandfather Vinay ripped up homes and displaced communities when building the Ugandan railway. Vinod now reflected on his father's complex legacy and motivations. Despite their troubled relationship, he empathized with Vinay's desire for security and belonging. Vinod realized he himself had fought hard to establish a home in Canada, sometimes neglecting to appreciate what he had. Now physically weakened but gaining perspective, Vinod hoped to find contentment. |
| | Hari went to the protest rally downtown with his friend Solomon and others. As the peaceful protest ended, tensions rose between protesters and police. Chaos broke out, with violence erupting on both sides. Police barricaded the protesters, using tear gas and force to disperse them. In the confusion, Hari lost sight of Solomon and the others. Blinded by the tear gas, Hari desperately tried to find his way and reconnect with his friends during the fray. |
| | Rajni and Vinod went searching for Hari during the chaotic protest. In their confusion amidst the clashes, they struggled to find their way home. Fortunately their daughter Meetu arrived and helped guide them back, reuniting with the rest of the family. They returned home to find their shop had been destroyed in a fire during the fray. Though devastated, the family drew strength from one another, committed to rebuilding together. |

Table 13: Claude 88000 summaries of Janika Oz's *A History of Burning*.

| Strategy | Summary |
|---|---|
| Hier | In 1972 in Uganda, Latika and Arun, an Indian couple with a baby son named Harilal, receive a mysterious letter delivered by a young Ugandan soldier. The country is under the dictatorship of Idi Amin, who has ordered all Asians to leave Uganda within 90 days. Latika hides the letter from Arun and reads it alone, discovering it is an order from Amin to immediately stop publishing their underground newspaper Jicho, which has been printing anti-government articles. Latika has secretly kept the paper going against Arun's wishes, as he argues it endangers their family while she believes justice is more important than their personal safety. |
| | Latika hides the letter out of fear of Arun's reaction and thrill at the power it places in her hands, as the soldier assumed Arun was in charge, not realizing she runs Jicho. Her secrecy over the paper and letter signals a rupture in trust between them, as Arun wants to protect his family but Latika is singleminded in her activism, sparking conflict in their marriage. With the expulsion order, this clash between family and justice becomes increasingly fraught. |
| | The story then jumps ahead to 1992 in Canada, where Rajni and Vinod are searching at night for their missing son Hari after a protest rally turned violent. They find their daughters Mayuri, Kiya and granddaughter Meetu waiting, but Hari is still missing. Earlier that day, Hari went to the rally despite his family's concerns. His friend Solomon was also there. When police arrived, tear gas was released and protesters fled as confrontations broke out. Hari lost Solomon in the chaos. |
| | Unable to find Hari, Rajni realizes they should go home, as home is where her family is, not any physical place. Just then, Mayuri spots them and brings them to their shop, which has been set on fire during the unrest. As the family watches the burning shop, Hari appears, having made his way back alone. Angry they are just standing there, he wants to try saving the shop, but Rajni says calmly it's done; they know how to rebuild. Vinod agrees, saying now is the time to rest. |
| | The fire destroys the shop but leaves the ground ready for something new. Hari realizes his family has persevered, almost whole. Though they will have to start over with the shop gone, they will insist on something better next time rather than accept injustice. Rajni reflects on how they have migrated from India to Africa to Canada, realizing anywhere can feel like home if her family is together. After so much loss, they made it here united. While the shop is destroyed, their family remains strong. |
| | In summary, the story spans 20 years and two families, exploring how broader political conflicts intertwine with and impact family relationships. In 1970s Uganda, rising political tensions strain the marriage of activists Latika and Arun as they clash over family versus justice. Two decades later in 1990s Canada, the family of Rajni and Vinod draw closer amidst violent unrest, losing their shop but keeping their bond intact. Despite different places and contexts, both families experience the complex dynamics between individual ideals, family ties, and turbulent political realities. Ultimately they share the resilient hope that as long as their family is together, they have arrived home. |
| Inc | In 1972, Latika and her husband Arun live in Kampala, Uganda with their baby son Harilal. Latika runs an underground newspaper called Jicho with her friend Daniel, who has recently disappeared likely due to being critical of dictator Idi Amin's regime. On the morning of August 4th, Amin declares over the radio that Asians must leave Uganda within 90 days. Latika's family is in shock, trying to understand if they will be impacted as Ugandan citizens. Her sister Mayuri is away studying medicine in India. Latika's father Vinod goes to pray at the temple and sees a dead body in the river, then finds a copy of Jicho which he destroys out of fear. |
| | The family spends days waiting in endless lines to try to get paperwork to leave, selling possessions, and being humiliated by soldiers. Latika's parents are unable to secure clearance to leave yet. One day, soldiers line up and shoot Asians on the street, but Vinod is spared because he has a red mark on his forehead from praying. He sees his childhood friend Moses, now a soldier, who lets him go. |
| | Meanwhile, a soldier delivers a letter to Latika ordering her to cease publication of Jicho, but she hides it from Arun. As the deadline to leave looms, Latika reflects on her tense relationship with Arun over her activism and thinks about why she chose him. She resolves to continue publishing the banned paper in secret, seeing it as the purpose of her life. On the night of September 9th, the eve of the expulsion deadline, soldiers break into Latika and Arun's apartment. They beat and arrest Arun while Latika screams and clings to their baby Harilal. The next day, Latika's parents and sister Kiya come to collect Harilal so they can flee the country together. But despite their pleas, Latika refuses to leave Kampala without Arun, entrusting them with caring for her son. Heartbroken, the family leaves Latika behind as they make their way to the airport and eventually resettle as refugees in Canada. |
| | Now alone in Kampala, Latika descends into despair searching for Arun. She publishes one final defiant issue of Jicho before her supplies run out. An old friend named Adroa visits and is shocked to find Latika still there. Seeing the toll it has taken on her, he encourages Latika to leave Uganda and make a new life elsewhere. Latika finally resolves to do so, boarding a plane to London with the other remaining Asian refugees. |

Table 14: Mixtral 2048 summaries of Janika Oz's *A History of Burning*.

| Strategy | Summary |
|---|---|
| Hier | "A History of Burning" by Janika Oza is an intricate, multigenerational saga that explores themes of intergenerational trauma, resilience, identity, and colonial exploitation across time, spaces, and continents. The narrative spans from Jamnagar, Gujarat, to Kenya, Uganda, and Canada, weaving together the lives of several characters whose experiences intersect and diverge amidst political upheaval, personal sacrifices, and cultural expectations. 

 The story begins in Jamnagar during a severe drought, where thirteen-year-old Pirbhai struggles to find work. Driven by his mother's encouragement, he embarks on a journey to Jamnagar and meets a mysterious merchant who leads him to work on a dhow sailing towards an unknown destination. Meanwhile, in Kenya, Sonal is a young Indian woman living with her family. When Pirbhai, now a refugee in Kenya, is employed by Sonal's father, the two form a deep bond as they navigate their lives together amidst political turmoil and personal sacrifices. 

 Pirbhai's past unveils his harrowing experiences as an indentured laborer in East African colonies and the guilt he feels over the circumstances that led him to Kenya. Simultaneously, the narrative explores Sonal's losses, including her brothers who perished during migration, and her struggle to forge connections with her family and her new home. 

 The narrative intertwines with Rajni's story during her migration from Uganda to Canada, following Idi Amin's expulsion of Asians from the country. Rajni, a resilient woman, grapples with her new life in a cold and foreign country while mourning the loss of her daughter, Latika. The lives of these characters continue to intersect and diverge in Toronto, revealing a rich tapestry of experiences, hopes, and triumphs despite the adversities they encounter. 

 Other characters play significant roles in shaping the narrative. Vinod, an Indian immigrant, navigates cultural assimilation, identity, and family expectations in Canada. Kiya, Rajni's daughter, deals with her resentment towards her sister Mayuri, who left the family during their time of need. Hari, the Patels' son, and his friend Solomon grapple with the consequences of their actions during a protest against racism. 

 Through the characters' experiences, the narrative explores the intricate connections between historical events and their impact on the characters' lives. From the struggles of poverty in Jamnagar, the brutal labor conditions in East African colonies, to the racial tensions and political turmoil in Uganda, the story emphasizes the power of resilience, human connections, and the complex nature of identity forged amidst shifting political alliances and dislocations. 

 In Toronto, Kiya joins the army, and the family grieves after Latika's disappearance. Hari and Solomon confront the consequences of their actions during a protest against racism. The non-linear storytelling features past and present flashbacks, providing a rich context for the characters' experiences. Despite the non-linear narrative, the story ultimately presents a consistent and chronological account of the characters' lives and their struggles to forge a sense of belonging amidst adversity. 

 In conclusion, "A History of Burning" is a compelling exploration of characters' lives woven together throughout history, emphasizing the profound power of resilience, human connections, and the complex nature of identity. By engaging with themes such as colonial exploitation, community, survival, and identity, the story forges a rich and imaginative world that touches on the lives of Pirbhai, Sonal, Rajni, Vinod, Kiya, Mayuri, and others, as they navigate across continents and navigate an intricate web of cultural expectations, personal sacrifices, and political upheaval. |
| Inc | The story follows the Pirbhai family, whose roots lie in India but have ties to Kenya, Uganda, and now Canada. The family includes Rajni and Vinod, their sons Hari and the late Arun, daughter Mayuri, Rajni's uncle Nanu, reemerged sister Kiya, and her son Harilal. In 1991, Hari bonds with friends Marcus, Solomon, and Sami over shared experiences and basketball. He grapples with Arun's death and the revelation of a long-lost sister, Latika. The family unites to discuss Latika and commemorate Arun. Hari investigates the Ugandan Asian expulsion but discovers limited information. 

 In 1992, Hari turns 19 and forms a close friendship with Marcus, Solomon, and Sami. Ignited by the Rodney King beating and the deaths of Raymond Lawrence and Michael Wade Lawson, he joins community protests, driven by his pursuit of justice and skepticism of official narratives. Together, they restore a vandalized local basketball court. Hari's reflections on family dynamics lead him to play basketball with Solomon instead of completing a report. 

 Parallelly, Rajni navigates Vinod's deteriorating health while embracing his desire for privacy. Together, they face his physical decline. Rajni's past reemerges while she raises her concerns for Hari's safety during protests. She understands the importance of supporting their causes despite her fears and apprehensions. 

 At the demonstration, Hari advocates for justice and unity. When tensions escalate, he gets separated from his friends. Rajni turns to prayer for Hari's safety, prompted by reminiscences of her past. Upon discovering Hari's disappearance, the family unites in searching for him. 

 After the protest, disoriented and lost, Hari reconnects with his friends, who assist him in returning home. Aware of Hari's safety, the family reconvenes to make sense of the day's events, solidifying their unity and commitment to fighting for justice. 

 The narrative intertwines Rajni, Hari, and their friendships as they confront their histories, find comfort in unity, and sustain their roots amid political strife. Acclaimed writers like Sharon Bala, Ngig) wa Thiong'o, and Gaiutra Bahadur enrich the storyline with dedicated support, historical context, and scholarship. Mentors and writing programs bolstered Hari's belief in his writing abilities. 

 The Pirbhai family's story spans generations and continents, with each member experiencing their own trials and triumphs. They carry the weight of their family legacy and the history of their ancestors. Through it all, the family remains united, finding strength in their shared experiences and their commitment to fighting for justice and equality. The story emphasizes the importance of community, collectivism, and solidarity, inviting readers to reflect on who is included and excluded in their own spheres. |

Table 15: LLaMA 2 2048 summaries of Janika Oz's *A History of Burning*.

| Strategy | Summary |
|---|---|
| Hier | In 1898, a young boy named Pirbhai leaves his home in India in search of work. He meets a merchant who offers him a job on a boat heading to an unknown destination. Along with several other men, Pirbhai embarks on the journey. In the late nineteenth century, Pirbhai leaves India to find work on a ship bound for Africa. Once aboard, he meets Jameel, a fellow countryman, and together they endure the hardships of life at sea. After several weeks, the ship arrives in Mombasa, Kenya, and Pirbhai and Jameel are among those who disembark. They are taken to a nearby railway camp, where they are put to work building a railway line through the jungle. Pirbhai struggles to adjust to life in the camp, where he is plagued by memories of his family back home and the guilt of being unaware of the true nature of his employment. Despite these challenges, he finds solace in the company of his fellow workers and in the simple pleasures of life in the camp. |
| | In 1956, Pirbhai and Rakesh work as coolies in India under the British Empire. They are tasked with burning down abandoned huts in the countryside, and Pirbhai feels conflicted about participating in such violence. In 1956, the day their third granddaughter Kiya was born, they find a portrait of Queen Elizabeth scattered in the street. They were on their way to make a donation at the local Asian school, whose roof was rusted and collected magpie droppings. |
| | Mayuri, a young Ugandan woman, receives a call from her friend Kiya in 1990. |
| | Pirbhai leaves India in search of work in 1898. He meets a merchant who offers him a job on a boat heading to an unknown destination. Along with several other men, Pirbhai embarks on the journey. In the late nineteenth century, Pirbhai leaves India to find work on a ship bound for Africa. Once aboard, he meets Jameel, a fellow countryman, and together they endure the hardships of life at sea. After several weeks, the ship arrives in Mombasa, Kenya, and Pirbhai and Jameel are among those who disembark. They are taken to a nearby railway camp, where they are put to work building a railway line through the jungle. Pirbhai struggles to adjust to life in the camp, where he is plagued by memories of his family back home and the guilt of being unaware of the true nature of his employment. Despite these challenges, he finds solace in the company of his fellow workers and in the simple pleasures of life in the camp. |
| | In 1956, Pirbhai and Rakesh work as coolies in India under the British Empire. They are tasked with burning down abandoned huts in the countryside, and Pirbhai feels conflicted about participating in such violence. In 1956, the day their third granddaughter Kiya was born, they find a portrait of Queen Elizabeth scattered in the street. They were on their way to make a donation at the local Asian school, whose roof was rusted and collected magpie droppings. |
| | Mayuri, a young Ugandan woman, receives a call from her friend Kiya in 1990. |
| | Pirbhai leaves India in search of work in 1898. He meets a merchant who offers him a job on a boat heading to an unknown destination. Along with several other men, Pirbhai embarks on the journey. In the late nineteenth century, Pirbhai leaves India to find work on a ship bound for Africa. Once aboard, he meets Jameel, a fellow countryman, and together they endure the hardships of life at sea. |

# M PROMPTS

## M.1 INCREMENTAL UPDATING

**Generate initial summary**

```
Below is the beginning part of a story:

---

{}

---

We are going over segments of a story sequentially to gradually
update one comprehensive summary of the entire plot. Write a
summary for the excerpt provided above, make sure to include
vital information related to key events, backgrounds, settings,
characters, their objectives, and motivations. You must briefly
introduce characters, places, and other major elements if they
are being mentioned for the first time in the summary. The story
may feature non-linear narratives, flashbacks, switches between
alternate worlds or viewpoints, etc. Therefore, you should
organize the summary so it presents a consistent and chronological
narrative. Despite this step-by-step process of updating the
summary, you need to create a summary that seems as though it is
written in one go. The summary should roughly contain {} words and
could include multiple paragraphs.

Summary ({} words):
```

**Generate intermediate summaries**

```
Below is a segment from a story:

---

{}

---

Below is a summary of the story up until this point:

---

{}

---

We are going over segments of a story sequentially to gradually
update one comprehensive summary of the entire plot. You are
required to update the summary to incorporate any new vital
information in the current excerpt. This information may relate to
key events, backgrounds, settings, characters, their objectives,
and motivations. You must briefly introduce characters, places,
and other major elements if they are being mentioned for the
first time in the summary. The story may feature non-linear
narratives, flashbacks, switches between alternate worlds or
viewpoints, etc. Therefore, you should organize the summary so
```

it presents a consistent and chronological narrative. Despite this step-by-step process of updating the summary, you need to create a summary that seems as though it is written in one go. The updated summary should roughly contain {} words and could include multiple paragraphs.

Updated summary ({} words):

**Compression**

Below is a summary of part of a story:

---

{}

---

Currently, this summary contains {} words. Your task is to condense it to less than {} words. The condensed summary should remain clear, overarching, and fluid while being brief. Whenever feasible, maintain details about key events, backgrounds, settings, characters, their objectives, and motivations – but express these elements more succinctly. Make sure to provide a brief introduction to characters, places, and other major components during their first mention in the condensed summary. Remove insignificant details that do not add much to the overall story line. The story may feature non-linear narratives, flashbacks, switches between alternate worlds or viewpoints, etc. Therefore, you should organize the summary so it presents a consistent and chronological narrative.

Condensed summary (to be within {} words):

## M.2 HIERARCHICAL MERGING

**Generate lowest-level summaries**

Below is a part of a story:

---

{}

---

We are creating one comprehensive summary for the story by recursively merging summaries of its chunks. Now, write a summary for the excerpt provided above, make sure to include vital information related to key events, backgrounds, settings, characters, their objectives, and motivations. You must briefly introduce characters, places, and other major elements if they are being mentioned for the first time in the summary. The story may feature non-linear narratives, flashbacks, switches between alternate worlds or viewpoints, etc. Therefore, you should organize the summary so it presents a consistent and chronological narrative. Despite this recursive merging process, you need to create a summary that seems as though it is written in one go.

The summary must be within {} words and could include multiple paragraphs.

Summary:

**Merge summaries**

Below are several summaries of consecutive parts of a story:

---

{}

---

We are creating one comprehensive summary for the story by recursively merging summaries of its chunks. Now, merge the given summaries into one single summary, make sure to include vital information related to key events, backgrounds, settings, characters, their objectives, and motivations. You must briefly introduce characters, places, and other major elements if they are being mentioned for the first time in the summary. The story may feature non-linear narratives, flashbacks, switches between alternate worlds or viewpoints, etc. Therefore, you should organize the summary so it presents a consistent and chronological narrative. Despite this recursive merging process, you need to create a summary that seems as though it is written in one go. The summary must be within {} words and could include multiple paragraphs.

Summary:

**Merge summaries with prior context**

Below is a summary of the context preceding some parts of a story:

---

{}

---

Below are several summaries of consecutive parts of a story:

---

{}

---

We are creating one comprehensive summary for the story by recursively merging summaries of its chunks. Now, merge the preceding context and the summaries into one single summary, make sure to include vital information related to key events, backgrounds, settings, characters, their objectives, and motivations. You must briefly introduce characters, places, and other major elements if they are being mentioned for the first time in the summary. The story may feature non-linear narratives, flashbacks, switches between alternate worlds or viewpoints,

```
etc. Therefore, you should organize the summary so it presents
a consistent and chronological narrative. Despite this recursive
merging process, you need to create a summary that seems as though
it is written in one go. The summary must be within {} words and
could include multiple paragraphs.
```

```
Summary:
```

### M.2.1  LLAMA 2 PROMPTS

**Generate lowest-level summaries**

```
Below is a part of a story:
```

```
---
```

```
{}
```

```
---
```

```
Write a coherent and chronological summary for the excerpt
provided above. Briefly introduce characters, places, and other
major elements if they are being mentioned for the first time in
the summary. The summary must be within {} words and could include
multiple paragraphs.
```

```
Summary:
```

**Merge summaries**

```
Below are several summaries of consecutive parts of a story:
```

```
---
```

```
{}
```

```
---
```

```
Merge the given summaries into one coherent and chronological
summary. Briefly introduce characters, places, and other major
elements if they are being mentioned for the first time in the
summary. The summary must be within {} words and could include
multiple paragraphs.
```

```
Summary:
```

**Merge summaries with prior context**

```
Below is a summary of the context preceding some parts of a story:
```

```
---
```

```
{}
```

```
---
```

```
Below are several summaries of consecutive parts of a story:
```

```
---
```

```
{}
```

---

```
Merge the preceding context and the summaries into one coherent
and chronological summary. Briefly introduce characters, places,
and other major elements if they are being mentioned for the first
time in the summary. The summary must be within {} words and could
include multiple paragraphs.
```

```
Summary:
```

## M.3   ARTIFACT REMOVAL

```
Below is a summary of a book:
```

---

```
{}
```

---

```
Your task is to edit the book summary by removing any phrases
that indicate it was developed progressively. Delete terms such
as "in the ... segment," "in ... part of the story," "in the ...
excerpt," "in the updated summary," and any similar phrases.
The goal is to make the summary read as if it was written all
at once, not in stages. In addition, eliminate any elements
taken from non-narrative sections like the table of contents,
acknowledgments, author's biography, author's note, information of
the author's other works, and so on. Apart from these adjustments,
do not make any other changes to the summary.
```

## M.4   GPT-4 AUTOMATIC EVALUATION

We use ellipsis here to keep this prompt concise. The complete version includes two full summaries and 42 sentence-level annotations, and will be made available in our codebase.

```
Given a book summary and a sentence from that summary, determine
if that sentence causes any confusion. Types of confusion include
the following:

- Entity omission: an entity, real or abstract (person, object,
place, concept, etc.) is mentioned, but key details are missing or
unclear
- Event omission: an event is mentioned, but key details are
missing or unclear
- Causal omission: the reason or motivation for something is
missing or unclear
- Salience: inclusion of trivial details that do not contribute to
the main storyline
- Discontinuity: an interruption in the flow of the narrative,
including but not restricted to: sudden jumps between
perspectives, time periods, or settings; poor transition between
sentences or paragraphs; sentences or paragraphs that seem out of
place; illogical sentence order or summary structure
- Duplication: redundant repetition of similar information
```

- Inconsistency: two parts of the summary contain contradicting information
- Language: grammar issues; confusing wording or phrasing; etc.

For something to qualify as a confusion, it must meet these two conditions:

1. Without resolving the confusion, readers would struggle substantially to grasp the main narrative, or the summary would appear incoherent.
2. The confusion can't be resolved solely using the information provided in the summary.

If the given sentence involves confusion that meets these two qualifications, ask relevant clarifying questions and provide the confusion types. There can be multiple questions, and keep in mind that a sentence may involve multiple types of confusion. If the given sentence doesn't involve any confusion, simply say "no confusion". Some examples are provided below, followed by the summary and sentence that you need to annotate.

---

[Example summary 1]

In the small town of Elm Avenue, teenage twins Aida and Salma are inseparable. When Aida mysteriously disappears, Salma recounts their childhood and the search for her sister begins...

...

...Throughout their time together, Sara and Juan explore their pasts, finding a unique bond and sense of comfort in each other's company, as the story moves through its diverse and interconnected characters and settings.

[Example sentence]
In the small town of Elm Avenue, teenage twins Aida and Salma are inseparable.

[Example response]
Questions: no confusion
Types: no confusion

[Example sentence]
When Aida mysteriously disappears, Salma recounts their childhood and the search for her sister begins.

[Example response]
Questions: no confusion
Types: no confusion

[Example sentence]
Meanwhile, the focus shifts to a new character, Fausto, and his relationship with Paz, as they navigate life in a hurricane-ravaged Miami neighborhood.

```
[Example response]
Questions: Why are we suddenly switching to a new character? Does
Aida have any connections with Fausto?
Types: discontinuity

...

---

[Example summary 2]

...

[Example sentence]
Proctor Bennett, a ferryman in Prospera, assists people
transitioning to new lives upon retirement.

[Example response]
Questions: Where or what is Prospera?
Types: entity omission

[Example sentence]
Struggling with dreams and discontent, his wife Elise suggests a
change in his work.

[Example response]
Questions: no confusion
Types: no confusion

[Example sentence]
Deciding to quit being a ferryman, Proctor teaches Caeli how to
swim.

[Example response]
Questions: How does quitting being a ferryman relate to teaching a
girl how to swim?
Types: discontinuity, causal omission

...

---

[Your summary]

{}

[Your sentence]

{}

[Your response] Determine if it the sentence above involves
confusion that can't be clarified using information from any part
```

of the given summary, and those which, if left unresolved, would
make the summary highly incoherent or significantly hinder readers
from understanding the main storyline. If you don't identify any
confusion like that, simply say "no confusion" for both questions
and types in your response.

