# OpenReview forum: "BooookScore: A systematic exploration of book-length summarization in the era of LLMs"
_ICLR.cc/2024/Conference — ICLR 2024 oral_

### Official Review · Reviewer_vK8q · 2023-10-28

**Soundness:** 3 good
**Presentation:** 3 good
**Contribution:** 3 good
**Rating:** 8
**Confidence:** 4

**Summary:**

The paper investigates book-length summarization. It proposes an automatic, LLM-based, reference- and source-free metric to evaluate the coherence of a summary. Using this metric, two different book-length summarization techniques are evaluated, and the evaluation is compared to human evaluation.

**Strengths:**

- The proposed metric is reference- and source-free, and thus has broad applicability. It also follows recently proposed best practices.
- To reduce the likelihood that LLMs have seen this data before, the paper used an evaluation dataset based on recently published books.
- The authors promise to release code and annotations.

**Weaknesses:**

- The evaluation of human annotations focuses only on precision and does not investigate recall. However, the authors are open about this limitation.
- Some of the examples of types of errors shown in Table 1 seem to be of questionable quality; for example, the first example about the "mysterious man" might as well be unanswered in the book (although I do not know the contents of the book in question). This may be a side effect of offering a monetary reward based in part on the number of annotations (cf. Appendix G).
- The sentence-level score may disproportionately favor summaries that contain a large number of (short) sentences. Furthermore, the evaluation of different models does not further investigate whether some of the score differences can be explained by the different length of the summaries (e.g., a shorter summary may be more prone to omissions).

**Questions:**

- Table 2: Why is there no entry for LLaMA2 in the incremental update section?
- Page 7, Section 5: Doesn't the choice of $p=1$ for nucleus sampling disable it completely? The choice of a large temperature also seems rather unusual for experimental evaluation, where a temperature of 0 is often chosen for reproducibility. Considering that Claude used a different temperature than other models, have you investigated how stable the results are when the temperature is varied?
- Page 7, Section 5, "Incremental..." paragraph - shouldn't "lower" by "higher"?
- Page 8, top paragraph: "Highly extractive" seems to contradict the large proportion of novel trigrams, which would be more indicative of an abstractive summary.

---

> ### Author Response · Authors · 2023-11-16
>
> We thank the reviewer for their insightful comments and for appreciating the applicability and other strengths of our work.
>
> *> The evaluation of human annotations focuses only on precision and does not investigate recall. However, the authors are open about this limitation.*
>
> In order to measure recall, we would require ground truth annotations with all possible coherence errors in the summaries. However, prior studies (SNaC and Scarecrow) have discussed that this is challenging for subjective fine-grained evaluation tasks where annotators generally display high precision but low recall.
>
> One solution could be to collect annotations from multiple annotations per summary, and consider their union as the exhaustive set of errors. Scarecrow and SNaC employ 10 and 3 annotators per summary respectively, but such a scheme would be very costly for our setting, as annotating 100 summaries already costs $1.5K. Annotation precision is more economical to evaluate, while also serving as an informative metric. We have updated Section 3 to make it clearer. We believe future work in human evaluation that makes recall-centric annotation more tractable is a promising area that could complement the precision-centric measurement we propose here.
>
> *> Some of the examples of types of errors shown in Table 1 seem to be of questionable quality…*
>
> This is a valid concern, and we appreciate the reviewer for pointing it out. We acknowledge that sometimes annotators or GPT-4 may ask questions that are also unanswered in the book, and there is no way to confirm without reading the source text. Using the reviewer’s example, if the identity of the mysterious man is not revealed in the book, the summary could say something like “A mysterious man whose identity is never revealed.” Since our focus is on coherence, the nature of our evaluation protocol is to have readers naturally raise questions as they read through a summary. Checking whether a question can be answered by the source text and finding answers to these questions would fall under faithfulness evaluation, which we leave to future work.
>
> *> This may be a side effect of offering a monetary reward based in part on the number of annotations (cf. Appendix G).*
>
> We appreciate the reviewer for going through the appendix. For the first evaluation task where we collected span-question pairs, we actually paid $15 USD per summary regardless of how many annotations were provided as mentioned in Section 3.
>
> Appendix G (now J) is for the second human evaluation where we validated annotations collected from the first one. For this evaluation, we paid based on how many annotations from the first task an annotator had to annotate, which also did not incentivize them in any way. The annotators were informed of the second evaluation task a few weeks after they completed the first one, so payment for the second task could not have biased annotators to create more annotations for the first. We understand that this may be unclear, and have made clarifications in Appendix J.
>
> *> The sentence-level score may disproportionately favor summaries that contain a large number of (short) sentences.*
>
> We understand this concern. In practice, we do not observe significant differences in sentence length. We computed the average number of tokens per sentence for the 100 summaries collected under each configuration (10 in total). The minimum is 21, maximum is 28.3, and standard deviation is 2.5. Future work may decompose the summary into atomic units (like FactScore) before running fine-grained evaluation. However, this decomposition may be hard to automate.
>
> *> Furthermore, the evaluation of different models does not further investigate whether some of the score differences can be explained by the different length of the summaries.*
>
> This is a valid concern. We have added a plot of BOOOOKSCORE vs. summary length in Appendix F, which shows no correlation between the two.
>
> *> Table 2: Why is there no entry for LLaMA2 in the incremental update section?*
>
> This is explained in Section 5, we could not get LLaMA 2 to perform incremental updating.
>
> *> Doesn't the choice of p = 1 for nucleus sampling disable it completely?*
>
> You are correct, we have rephrased it in the paper.
>
> *> The choice of a large temperature also seems rather unusual…have you investigated how stable the results are when the temperature is varied?*
>
> We found that setting temperature to 0 affected generation quality in our preliminary experiments. Plus, prior to the recent OpenAI update, setting temperature to 0 for these closed-source models did not guarantee deterministic output. We did not study the effect of temperature on BOOOOKSCORE due to budget constraints.
>
> *> shouldn't "lower" be "higher"?*
>
> That is correct, we have changed it in the paper.
>
> *> “Highly extractive" seems to contradict the large proportion of novel trigrams…*
>
> “Highly extractive” is relative to other models. We have updated the paper with clarification.

---

> > ### Comment · Reviewer_vK8q · 2023-11-16
> >
> > First of all, thank you for your concise response and the structure that makes it easy to follow each point. I will comment on a few specific points below, but overall my concerns have been addressed. I will update my rating accordingly.
> >
> > >>_Some of the examples of types of errors shown in Table 1 seem to be of questionable quality…_
> > >
> > >This is a valid concern, and we appreciate the reviewer for pointing it out. We acknowledge that sometimes annotators or GPT-4 may ask questions that are also unanswered in the book, and there is no way to confirm without reading the source text. Using the reviewer’s example, if the identity of the mysterious man is not revealed in the book, the summary could say something like “A mysterious man whose identity is never revealed.” Since our focus is on coherence, the nature of our evaluation protocol is to have readers naturally raise questions as they read through a summary. Checking whether a question can be answered by the source text and finding answers to these questions would fall under faithfulness evaluation, which we leave to future work.
> >
> > Thanks for your clarification. I also read your response to reviewer `SAFf`'s question about faithfulness, including the unreasonable effort to collect such labels.
> >
> > >> _This may be a side effect of offering a monetary reward based in part on the number of annotations (cf. Appendix G)._
> > >
> > >We appreciate the reviewer for going through the appendix. For the first evaluation task where we collected span-question pairs, we actually paid $15 USD per summary regardless of how many annotations were provided as mentioned in Section 3.
> > >Appendix G (now J) is for the second human evaluation where we validated annotations collected from the first one. For this evaluation, we paid based on how many annotations from the first task an annotator had to annotate, which also did not incentivize them in any way. The annotators were informed of the second evaluation task a few weeks after they completed the first one, so payment for the second task could not have biased annotators to create more annotations for the first. We understand that this may be unclear, and have made clarifications in Appendix J.
> >
> > Thank you for your explanation, which clears up my doubts. I think your additions to the paper are helpful to the reader.
> >
> > >>_Furthermore, the evaluation of different models does not further investigate whether some of the score differences can be explained by the different length of the summaries._
> > >
> > >This is a valid concern. We have added a plot of BOOOOKSCORE vs. summary length in Appendix F, which shows no correlation between the two.
> >
> > Thank you for the clarification with additional results. They dispel my doubts on this point. For the sake of completeness, I would be interested in an analogous plot for hierarchical summaries, e.g. as differently colored dots in the same image, or a separate plot next to it, although I assume they paint a similar picture.
> >
> > >> _Table 2: Why is there no entry for LLaMA2 in the incremental update section?_
> > >
> > > This is explained in Section 5, we could not get LLaMA 2 to perform incremental updating.
> >
> > Thank you for the reminder. It looks like I didn't remember that part correctly.

---

### Official Review · Reviewer_SAFf · 2023-10-31

**Soundness:** 4 excellent
**Presentation:** 4 excellent
**Contribution:** 4 excellent
**Rating:** 10
**Confidence:** 5

**Summary:**

This paper tackles the problem of summarization of books. This is an ambitious task (that would have been considered unattainable a few years ago), as well as very expensive (because any annotation or even evaluation is very cognitive intense and long). Very little research is done in that area so far, and most of it is based on automatically scraped material.

**Strengths:**

There is much to like in this paper. It is a great idea, and has been very, very well executed.

- The authors obtain 100 recent books from the last year - while reviews of those books exist those are not summaries. I would be less certain that those books or their summaries are _impossible_ to be part of the training corpus. Modern LLM go well beyond only crawling Internet data and providers have deals with many different publishers. Still, this is arguably one of the most rigorous attempts to create a test set that avoids dataset contamination.

- The evaluation focuses on precision. The standard approach of evaluation summaries is to rely on human-written summaries and the compare those with system-generated. This is very expensive, and creates lots of pitfalls as it depends on the similarity metric used to compare summaries (even without using n-gram overlap metrics, learnt embedding metrics like BERTScore have their own set of biases). Instead, this work focuses on assessing a given summary by asking questions about it.

- It compares the two standard ways of summarizing very long documents: hierarchical (summarizing parts and combining those) and incremental (left to right) with interesting insights  (eg, incremental is preferred for level of detail, but not for other aspects)


This work will certainly be considered a fundamental paper in the coming months and years, and be a required reference and reading for summarization research. It jumps right into what is still difficult to do with modern LLMs

**Weaknesses:**

I have very little concerns about this paper. It is however unfortunate that the evaluation framework did not consider faithfulness. This could have been done by asking annotators to assess each fact and search for it inside the book to verify its factuality.

**Questions:**

How did you chose the number of `O` in `BOOOOKSCORE` ?

---

> ### Author Response · Authors · 2023-11-16
>
> We extend sincere gratitude to the reviewer for their insightful comments and for greatly appreciating the value and importance of our work.
>
> *> It is however unfortunate that the evaluation framework did not consider faithfulness. This could have been done by asking annotators to assess each fact and search for it inside the book to verify its factuality.*
>
> We thank the reviewer for proposing a possible experimental setup for evaluating faithfulness. Coincidentally, we did try the task ourselves using the exact setup as described in the reviewer’s comments, and found it incredibly challenging. Some facts may span multiple pages, even chapters, of a book, and verifying a single fact could take 10-20 minutes, let alone a full summary’s worth. Considering the amount of time, money, and energy it would take to complete a full-scale human evaluation of faithfulness, we decided to stick to coherence and acknowledge that our work is complementary in nature. Prior work like LongEval and FactScore have established guidelines and methods for evaluating faithfulness, and our evaluation protocols can always be adapted to or combined with these work for faithfulness evaluation. We focus on coherence as it emerges as a noteworthy problem on its own when it comes to evaluating very long summaries of book-length documents.
>
> *> How did you choose the number of O in BOOOOKSCORE?*
>
> There are 4 O’s in BOOOOKSCORE because we have four authors on the paper!

---

### Official Review · Reviewer_hJ5Z · 2023-11-01

**Soundness:** 3 good
**Presentation:** 4 excellent
**Contribution:** 4 excellent
**Rating:** 8
**Confidence:** 4

**Summary:**

This paper addresses the problem of book-length summarization evaluation, which refers to documents with 100k+ tokens. It identifies the following main challenges: 1) the problem of data contamination, that is, most books are already in the pre-training data of large language models; 2) the lack reliable automatic metrics. To solve the first issue, the authors collect a dataset set of 100 recently published books. To address the second issue, they generate summaries for each book with GPT-4, and then ask humans to identify and classify coherence errors such as "entity omission", "causal omission", and "duplication". Then, to automate this process, they prompt GPT-4 to classify sentences in summaries according to the same coherence error taxonomy, and they find the precision of GPT-4 is similar to the human annotators. The coherence annotation by GPT-4 is the basis of BooooKScore, a reference-free automatic metric counts the fraction of sentences in summaries that are free from coherence issues. Finally, the paper evaluates summaries generated by GPT-4, Claude 2, and LLama-2-7B using two techniques: hierarchical merging and incremental updating. They find that GPT-4 and Claude 2 summaries are more coherent and that hierarchical merging also results in more coherent summaries, but at a cost of detail.

**Strengths:**

The paper addresses an important problem in summarization, proposing solutions to scale the evaluation of very long documents that are longer (for now) than current context sizes of large language models. The effort applied in validating a reference-free automatic metric with newly collected books and human annotation are relevant contributions for future work in summarization.

**Weaknesses:**

One point I miss in the experiments is a baseline without hierarchical merging or incremental updating. The reason is that a significant fraction of books have length around 100k tokens or less (if we observe the statistics of BookSum, for instance), and it would be interesting to see if the hierarchical merging or incremental updating introduce (or not) a high quantity of coherence issues compared to a vanilla LLM approach. Even with some level of truncation, it should still be possible to assess coherence issues.

Minor issue: you mention in section 3 that "we did not find summaries online for any books in our dataset, which means LLM memorization of these summaries is impossible." Not finding results online by no means imply that memorization is impossible. In fact, we have no guarantee that closed-source models such as GPT-4 are trained just on publicly available data.

**Questions:**

In the description of "incremental updating" in section 2, you state that "One major issue with the hierarchical merging approach is that it necessitates summarizing portions of the input document without complete context ... which can lead to incoherent summaries especially for non-linear or multi-perspective narratives." However, your experimental findings show that hierarchical merging results in more coherent summaries. Do you have an explanation for this observation?

---

> ### Author Response · Authors · 2023-11-16
>
> We thank the reviewer for their insightful comments and for appreciating the importance of our work to future research in summarization.
>
> *> One point I miss in the experiments is a baseline without hierarchical merging or incremental updating.*
>
> This is indeed a valid concern, we appreciate the reviewer for bringing it up. To address this issue, we evaluated GPT-4 on the SQuALITY dataset, which contains short sci-fi stories that are 4000-6000 words long. Results from these experiments have been added to Appendix E. Here, we show that the baseline method (without merging or incremental updating) produces slightly better summaries than incremental updating (in terms of ROUGE-L score), and we are currently running BOOOOKSCORE on this dataset as well for a more specific analysis of coherence. Additionally, qualitative inspection of these summaries (added in Appendix E.1) shows that they are not only superior to those produced by incremental updating but also human reference summaries from the dataset, which is aligned with results from recent work on LLM-based short-document summarization: https://arxiv.org/abs/2309.09558. Our experiments overall suggest that approaches like incremental updating can introduce coherence errors into LLM-generated summaries.
>
> *> Not finding results online by no means imply that memorization is impossible.*
>
> It is true that closed-source models may have been trained on publicly available data, we thank the reviewer for pointing this out. We have adjusted our wording in the paper to be more precise. That said, there is still a very low chance that these models have seen actual summaries of those books given how recently the books were published. All books in our dataset were published after the beginning of 2022, and 87% of them were published in 2023. For our experiments, we used the gpt-4 2023-03-15 and gpt-3.5-turbo-0301 checkpoints on Microsoft Azure. Both models have a training data cutoff date of Sep. 2021. Anthropic unfortunately does not disclose checkpoint information, but our summaries were all obtained via their Claude 2 API in September 2023. We also note that most of the books are not (yet) widely read, making it even less likely that readers have written high-quality summaries. Overall, we made diligent efforts within our control to avoid testing on the training set.
>
> *> However, your experimental findings show that hierarchical merging results in more coherent summaries. Do you have an explanation for this observation?*
>
> Prior to running our experiments, our theory was that incremental updating should enable the model to take more context into consideration when composing summaries. We understand that our phrasing in the paper is too definitive, and have revised it to clarify that this was our hypothesis.
>
> Our results show the opposite, and we hypothesize some reasons for this in Section 5 of the paper. Essentially, incremental updating is inherently a more complex task, as it requires the model to maintain a running summary as it steps through chunks of a book. When deciding what to include from the current book chunk and what to remove from the current summary, the model needs to juggle texts of different levels of granularity. Plus, the model must compress the summary whenever it exceeds the final summary limit, which could further affect coherence. For the chunk size = 2048 setting, models need to compress 17.6 times per book on average, while each book has 68 chunks on average. Hierarchical merging, on the other hand, only requires the model to summarize individual chunks and recursively merge them together. If a single summary gets too long, it would often be enough to simply have the model regenerate. This approach does not involve working with texts of different granularity levels or compressing summaries while maintaining coherence.

---

> > ### Comment · Reviewer_hJ5Z · 2023-11-18
> >
> > Thank you for the clarifications and additional experiments! I still think it is important to have a baseline evaluated on longer books, but I understand they are costly to obtain. At the moment, there is Claude with 100k context length and GPT-4 with 128k, and your metric could provide some direction on how promising is to invest in longer contexts. I will adjust my assessment accordingly.

---

### Official Review · Reviewer_rPVA · 2023-11-08

**Soundness:** 4 excellent
**Presentation:** 4 excellent
**Contribution:** 4 excellent
**Rating:** 8
**Confidence:** 4

**Summary:**

This paper provides a systematic empirical survey of the coherence problem in book-length summarization with LLMs. The contributions include a novel evaluation protocol, an automatic metric for coherence assessment, and a systematic evaluation of different LLMs.

**Strengths:**

The paper presents a very systematic survey, with solid and extensive experiments. The writing is clear and the narrative is easy to follow, facilitating understanding of complex concepts. The creation of the BOOOOKSCORE metric is particularly innovative, offering a decent solution to the cost and time constraints of human evaluation.

**Weaknesses:**

A potential weakness lies in the design of the BOOOOKSCORE metric. The approach of weighing all sentences equally may overlook the varying importance of different parts of the text for overall coherence. Additionally, there is an absence of a consistency check for the evaluation metric.

**Questions:**

N/A

---

> ### Author Response · Authors · 2023-11-16
>
> We thank the reviewer for their insightful comments and for appreciating our extensive experiments, clear writing, and innovative evaluation metric design.
>
> *> The approach of weighing all sentences equally may overlook the varying importance of different parts of the text for overall coherence.*
>
> We acknowledge that weighing all sentences equally may overlook the relative importance of different parts of the text to overall coherence. One way to mitigate this problem is to assign severity weights to different error types, as in MQM for machine translation. However, this step introduces more subjectivity into the evaluation procedure (e.g., see https://arxiv.org/abs/2305.18201 for similar concerns in long-form QA), which is why we chose to equally weight all errors. It is unclear how the weights should be chosen, and whether a single weighting scheme makes sense in all scenarios. Future work can further explore correlations between different weighting schemes and human judgments of overall summary coherence.
>
> *> there is an absence of a consistency check for the evaluation metric.*
>
> We understand the importance of the consistency check (i.e., if you resampled summaries from each LLM, would you obtain the same ranking of systems?), and appreciate the reviewer for bringing up this issue. That said, running multiple iterations of full-scale experiments is expensive (it can cost roughly $1-2K to collect 100 summaries from closed-source LLMs, depending on which model and chunk size are used). Instead, we ran a bootstrapping analysis where given 100 summaries generated under one configuration, we generated 1000 random samples with replacement using a sample size of 100, then computed BOOOOKSCORE for the 1000 samples. The scores have a standard deviation of 0.015, which indicates that BOOOOKSCORE is stable across varying random samples. We have added this analysis in Appendix G.

---

### Meta-Review · Area_Chair_Ahyj · 2023-12-07

**Metareview:**

Good paper. Clear accept.

The remaining concerns that have not been addressed by the rebuttal (with all the reviewers agree that these do not hurt the acceptance):

- A potential weakness lies in the design of the BOOOOKSCORE metric. The approach of weighing all sentences equally may overlook the varying importance of different parts of the text for overall coherence.
- There is an absence of a consistency check for the BOOOOKSCORE metric.
- The evaluation of human annotations focuses only on precision and does not investigate recall. However, the authors are open about this limitation.

**Justification For Why Not Higher Score:**

- A potential weakness lies in the design of the BOOOOKSCORE metric. The approach of weighing all sentences equally may overlook the varying importance of different parts of the text for overall coherence.
- There is an absence of a consistency check for the BOOOOKSCORE metric.
- The evaluation of human annotations focuses only on precision and does not investigate recall. However, the authors are open about this limitation.

**Justification For Why Not Lower Score:**

I can find no reason to give a lower score.

---

### Decision · Program_Chairs · 2024-01-16

Accept (oral)